# New Hybrid Fine-Tuning Paradigm for LLMs: Algorithm Design and Convergence Analysis Framework

**Shaocong Ma**
Department of Computer Science
University of Maryland
College Park, MD 20742, USA
`scma0908@umd.edu`

**Peiran Yu**
Department of Computer Science and Engineering
University of Texas Arlington
Arlington, TX 76019, USA
`peiran.yu@uta.edu`

**Heng Huang**[*]
Department of Computer Science
University of Maryland
College Park, MD 20742, USA
`heng@umd.edu`

## Abstract

Fine-tuning Large Language Models (LLMs) typically involves either full fine-tuning, which updates all model parameters, or Parameter-Efficient Fine-Tuning (PEFT), which adjusts a small subset of parameters. However, both approaches have inherent limitations: full fine-tuning is computationally expensive, while PEFT often struggles to learn new knowledge and exhibits suboptimal performance. To overcome these issues, we propose a novel *hybrid fine-tuning* approach that jointly updates both LLMs and PEFT modules using a combination of zeroth-order and first-order optimization methods. To analyze our new algorithm, we develop a theoretical framework centered on the concept of *hybrid smoothness condition*, which accounts for the heterogeneous nature of the optimization landscape in joint LLM and PEFT training. We derive a rigorous convergence analysis for the convergence of reshuffling-type SGD algorithm under multiple learning rates and demonstrate its effectiveness through extensive empirical studies across various downstream tasks and model architectures. On the practical side, our results demonstrate consistent performance improvement, making the approach a viable solution for large-scale language model fine-tuning.

## 1 Introduction

Large Language Models (LLMs) have emerged as an important paradigm in natural language processing (NLP), demonstrating remarkable capabilities across a wide range of tasks. To adapt these models for specific domains or to modify their core behaviors, researchers commonly employ full fine-tuning for downstream tasks (Malladi et al., 2023; Zhang et al., 2024; VM et al., 2024; Minaee et al., 2024), which updates all parameters of an LLM. However, this method is extremely computationally expensive, requiring the calculation of gradients for the entire model. To address this limitation, two common approaches have introduced: (1) *Zeroth-order full fine-tuning* (Malladi et al., 2023; Zhang et al., 2024; Gautam et al., 2024; Tang et al., 2024; Wang et al., 2024; 2025): This type of methods approximates gradients without directly computing them, reducing computational overhead while still updating all model parameters. (2) *Parameter-Efficient Fine-Tuning (PEFT) methods* (Lester et al., 2021; Hu et al., 2021; Li & Liang, 2021): These techniques aim to adapt LLMs by tuning only a small portion of parameters while keeping the base model frozen.

---
[*]This work was partially supported by NSF IIS 2347592, 2348169, DBI 2405416, CCF 2348306, CNS 2347617, RISE 2536663.

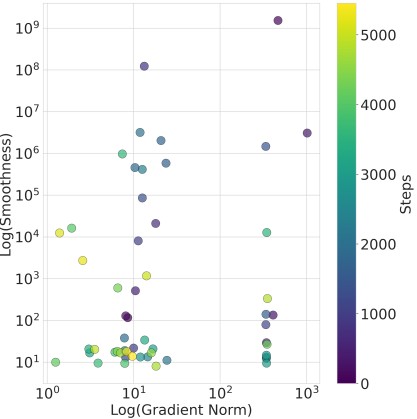
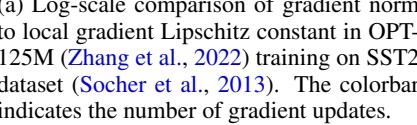
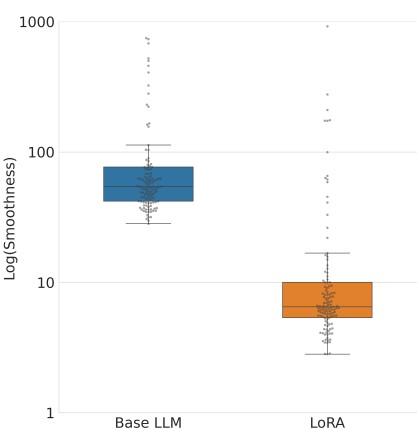

(a) Log-scale comparison of gradient norm to local gradient Lipschitz constant in OPT-125M (Zhang et al., 2022) training on SST2 dataset (Socher et al., 2013). The colorbar indicates the number of gradient updates.

(b) Comparison of gradient Lipschitz constant $L$ for different modules (OPT-125M and LoRA (Hu et al., 2021)). The base LLM exhibits a significantly larger, necessitating a smaller learning rate in gradient updating.

Figure 1: Visualization of smoothness structures in hybrid fine-tuning a large language model. These complex characteristics pose new challenges for the convergence analysis of traditional optimization algorithms, motivating us to consider a relaxed smoothness condition, *hybrid smoothness condition* (Definition 1), for the hybrid fine-tuning method.

However, directly applying either of these methods has been shown to be insufficient: As pointed out by (Gudibande et al., 2023) and (Ghosh et al., 2024), the PEFT method (*e.g.* LoRA) does not learn new knowledge, while the zeroth-order full fine-tuning suffers from slow convergence due to the lack of gradient information (Nesterov & Spokoiny, 2017). These limitations highlight a critical gap in current approaches, leading to the following question:

> ***Q1:*** *How can we achieve both benefits of full fine-tuning and PEFT methods while maintaining the efficiency?*

To address this question, we propose a novel approach, *hybrid fine-tuning*, which jointly updates both the PEFT module and the LLM. We integrate both first-order (FO) and zeroth-order (ZO) optimization techniques for conducting PEFT and updating the base model simultaneously. By leveraging ZO methods, we can perform fine-tuning on the base LLM without calculating the full gradient, thereby effectively learning new knowledge. Meanwhile, our method updates PEFT modules using the FO gradient information, speeding up traditional ZO full fine-tuning.

To assess efficiency, we analyze the convergence of our proposed approach, which presents new theoretical challenges in the analysis. As demonstrated in existing literature (Zhang et al., 2019; Carmon et al., 2020), the optimal learning rate is closely tied to the local smoothness of the loss landscape. This dependence is especially critical in our setting, where the complex architecture of modern LLMs and the heterogeneous nature of hybrid fine-tuning introduce **two key challenges**:

a) **A dynamic changing gradient Lipschitz constant.** The local smoothness structure of LLMs evolves dynamically during training. This phenomenon, first observed by (Zhang et al., 2019) for LSTM-based language models, extends to transformer-based architectures, underscoring the complexity of LLM fine-tuning. The Figure 1a illustrates this dynamic behavior in OPT-125M (Zhang et al., 2022), a transformer-based language model.

b) **Heterogeneous smoothness across parameters.** The base LLM and PEFT modules exhibit distinct smoothness characteristics. Due to differences in architecture and scale, components in our proposed hybrid fine-tuning approach naturally possess diverse smoothness properties. This heterogeneity is demonstrated in the Figure 1b, which compares the gradient Lipschitz constants between the base LLM and the LoRA module.

These challenges highlight a significant gap between existing theoretical frameworks and the practical implementation of hybrid fine-tuning methods: Traditional convergence analysis of optimization algorithms cannot be applicable for such complicated loss surface, which also leads to the following central question:

> **Q2:** *How can we develop a unified theoretical framework that accurately characterizes the convergence of SGD for hybrid fine-tuning while accounting for their distinct characteristics and behaviors?*

To answer this question, we develop a novel theoretical framework centered around the concept of *hybrid smoothness condition*. This framework provides a more accurate characterization of the optimization landscape in joint LLM and PEFT training, enabling rigorous analysis of convergence properties and optimization dynamics. Our main contributions are summarized as follows:

(1) We propose the hybrid fine-tuning paradigm, a novel approach that addresses the limitations of both full fine-tuning and traditional PEFT methods. By combining zeroth-order optimization for LLMs with first-order methods for PEFT modules, we achieve a balance between adaptation power and computational efficiency. This innovative strategy further reveals the *hybrid smoothness condition* (Definition 1) of the hybrid structure, highlighting a new theoretical challenge arising from the heterogeneous structure of joint LLM and PEFT optimization.

(2) To address the challenge posed by *hybrid smoothness condition*, we introduce a unified theoretical framework for analyzing hybrid optimization problems arising in hybrid fine-tuning. Within this framework, we establish the convergence of SGD with Random Reshuffling (Theorem 1), addressing a previously unresolved gap in optimization theory. Notably, our analysis extends the optimal sample complexity guarantees from the standard smooth loss class to the more general hybrid smooth loss function class.

(3) We conduct extensive empirical studies to evaluate the effectiveness of our hybrid fine-tuning approach across a diverse set of downstream tasks and model architectures. As shown in Figure 3 and Table 1, hybrid fine-tuning consistently outperforms existing methods across 18 model-task combinations (spanning three architectures and six tasks), achieving the highest accuracy in $94.5\%$ of the cases (17 out of 18). Empirical evidence of faster convergence is further validated in Figure 4. Notably, these improvements incur no additional memory overhead compared to the FO counterpart, as demonstrated in Table 2.

## 2 HYBRID FINE-TUNING AND HYBRID SMOOTHNESS CONDITION

### 2.1 OUR PROPOSED METHOD: THE HYBRID FINE-TUNING

To balance the adaptation power of full fine-tuning with the efficiency of PEFT, we introduce *hybrid fine-tuning*, where both the base LLM and a lightweight PEFT module are updated jointly.

**Methodology.** Our *hybrid fine-tuning* approach jointly updates both the PEFT module parameters and the base LLM parameters. The parameters updating tasks can be formulated as a class of optimization problems where the parameter space is partitioned into two distinct subspaces: Let $x \in \mathbb{R}^{d_x}$ denote the LLM parameters and $y \in \mathbb{R}^{d_y}$ the PEFT module parameters, $d = d_x + d_y$. For a dataset $\mathcal{D} = \{\xi_i\}_{i=1}^n$ we minimize the empirical loss:

$$\min_{(x,y) \in \mathbb{R}^d} f(x,y) := \frac{1}{n} \sum_{i=1}^n f(x,y;i). \tag{1}$$

In hybrid fine-tuning, we leverage ZO optimization for the $x$ parameters, which avoids computing the full gradient and thus significantly reduces memory requirements. Simultaneously, we update the much smaller PEFT module parameters (the $y$ parameter) using the FO gradient information, which leads to faster convergence and better performance compared to solely ZO methods. Our algorithm is described in Algorithm 1:

---

**Algorithm 1:** SGD with Random Reshuffling for Hybrid Fine-Tuning

---

**Input:** Learning rate $\eta = \begin{bmatrix} \eta_x & \eta_y \end{bmatrix}$, number of epochs $T$, dataset $\mathcal{D} = \{\xi_i\}_{i=1}^n$
Initialize the parameter at $(x_0, y_0)$;
**for** $t = 1$ **to** $T$ **do**
    Shuffle the dataset $\mathcal{D}$ to obtain $\mathcal{D}_t$;
    $x_{t,0}, y_{t,0} \leftarrow x_{t-1}, y_{t-1}$;
    **for** $i = 1$ **to** $n$ **do**
        $\begin{bmatrix} x_{t,i} \\ y_{t,i} \end{bmatrix} \leftarrow \begin{bmatrix} x_{t,i-1} \\ y_{t,i-1} \end{bmatrix} - \begin{bmatrix} \eta_x & 0 \\ 0 & \eta_y \end{bmatrix} \begin{bmatrix} \hat{\nabla}_x f(x_{t,i}, y_{t,i}; \xi_{t,i}) \\ \nabla_y f(x_{t,i}, y_{t,i}; \xi_{t,i}) \end{bmatrix}$;
    **end**
**end**
$x_t \leftarrow x_{t,n}$;
**Output:** Final parameters $x_T$

---

The optimization strategy is implemented using SGD with *random reshuffling*, a common practice in deep learning (Paszke et al., 2019; Abadi et al., 2016) demonstrated improved efficiency in existing theoretical literature (Ma & Zhou, 2020; Safran & Shamir, 2020; Mishchenko et al., 2020; Gürbüzbalaban et al., 2021; Liu & Zhou, 2024). Here, $\eta_x$ and $\eta_y$ are the learning rates for $x$ and $y$ parameters, respectively, $T$ is the total number of epochs, $\mathcal{D} = \{\xi_i\}_{i=1}^n$ is the dataset with $n$ samples, and $x_{t,i}$ and $y_{t,i}$ are the parameter values after the $i$-th iteration of the $t$-th epoch. $\hat{\nabla}_x f$ and $\nabla_y f$ are the stochastic gradients with respect to $x$ and $y$. Here, we use $\hat{\nabla}_x f$ to represent the gradient estimator of $\nabla_x f$. It is commonly estimated using the two-point gradient estimator defined as follows:

$$\hat{\nabla}_x f(x, y; \xi) := \frac{f(x + \mu v, y; \xi) - f(x, y; \xi)}{\mu} v, \tag{2}$$

where $v$ is a random Gaussian vector with identity covariance matrix (that is, $v \sim N(0, I_d)$) and $\mu$ is the perturbation stepsize.

## 2.2 CHALLENGES IN HYBRID FINE-TUNING: THE HYBRID SMOOTHNESS CONDITION

Besides intuitively designing the hybrid fine-tuning strategy, we conduct rigorous theoretical analysis on the convergence of Algorithm 1. However, the convergence analysis reveals **theoretical challenges** stemming from the complex optimization landscape of hybrid fine-tuning. Traditional analysis often relies on the $L$-smoothness assumption, which states that the gradient is Lipschitz continuous with a constant $L$, or equivalently, $\nabla^2 f(w) \preceq L I_d$. While $L$-smoothness has been demonstrated to hold for all smooth functions over a compact domain (Hewitt & Stromberg, 2012), this assumption is often too restrictive for deep learning models and particularly for our proposed hybrid setting. We recap these limitations we have introduced:

a) **The gradient Lipschitz constant $L$ is dynamically changing during training.** The constant $L$ usually fails to maintain uniformity over the entire parameter space. In many practical scenarios, different regions of the parameter space may exhibit vastly different smoothness properties. For instance, (Zhang et al., 2019) has demonstrated that the local smoothness constant $L$ is linear in the gradient norm. We also have illustrated this non-uniformity for transformer-based language models in Figure 1a.

b) **The gradient Lipschitz constant $L$ can be different for different parameters.** The base LLM (the $x$ parameter) and the PEFT module (the $y$ parameter) inherently possess different structural properties and scales. For example, small randomly-initialized modules often have smaller $L$ compared to large pre-trained neural networks. This consideration becomes particularly crucial in hybrid systems where we deal with fundamentally different types of parameters. We have illustrated this point in the Figure 1b: The LoRA module demonstrates a substantially lower $L$ value compared to the base LLM.

To rigorously characterize the phenomenon observed in hybrid fine-tuning, we adapt the concept of generalized smoothness (Zhang et al., 2019; Li et al., 2024) to our hybrid setting, leading to the *hybrid smoothness* condition:

**Definition 1** (Hybrid smoothness). *A function $f : \mathbb{R}^{d_x} \times \mathbb{R}^{d_y} \to \mathbb{R}$ has the hybrid generalized smoothness property if there exist two non-negative non-decreasing sub-quadratic functions $\ell_x : \mathbb{R}_{\geq 0} \to \mathbb{R}_{\geq 0}$ and $\ell_y : \mathbb{R}_{\geq 0} \to \mathbb{R}_{\geq 0}$ such that for all $(x, y)$:*

$$\begin{bmatrix} \ell_x(\|\nabla f(x,y)\|)I_{d_x} & 0 \\ 0 & \ell_y(\|\nabla f(x,y)\|)I_{d_y} \end{bmatrix} \succeq \nabla^2 f(x,y).$$

This definition allows the smoothness bound to depend on the current gradient norm and differ between the $x$ and $y$ parameter blocks. It naturally captures the dynamic and heterogeneous nature of the optimization landscape. It is easy to see that the standard $L$-smoothness is a special case where $\ell_x(t) = \ell_y(t) = L$ for all $t$. As demonstrated by (Zhang et al., 2019; Li et al., 2024), many neural network loss landscapes are empirically observed to be generalized smooth but not $L$-smooth.

**The Impact of Hybrid Smoothness Condition.** The generalized smoothness condition presents a significant challenge in training our proposed hybrid system, particularly motivating the use of two distinct learning rates. In the following example of our proposed hybrid LLM fine-tuning structure, we have two sets of parameters: (1) the original LLM parameters $x$, and (2) the PEFT module parameters $y$. we jointly train the LLM with a Prompt Encoder (Lester et al., 2021) on the SST-2 dataset (Socher et al., 2013). We observe that the base LLM merely takes much smaller learning rate; if we choose the learning rate to ensure the base LLM's convergence, the training loss decreases in an unacceptably slow rate (Figure 2a). However, if we choose the learning rate larger than the base LLM's tolerance, the training loss explodes and quickly diverges (Figure 2b). The best practice is choosing a smaller learning rate for the base LLM and a larger learning rate for the PEFT module (Figure 2c).

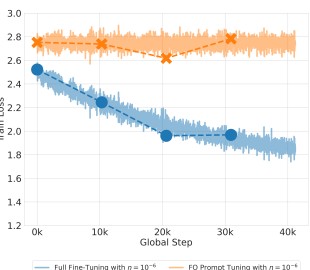 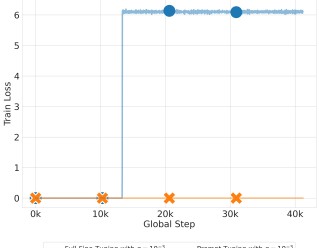 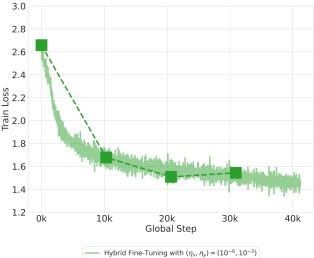

(a) Under the small learning rate $\eta = 10^{-6}$, the full fine-tuning decrease as expected. However, prompt tuning converges slowly and stagnates at a higher loss.

(b) With a large learning rate $\eta = 10^{-3}$, the prompt tuning decreases as expected, while full fine-tuning exhibits unstable behavior, resulting in loss explosion.

(c) Hybrid fine-tuning with distinct learning rates ($\eta_x = 10^{-6}$ for the base model, $\eta_y = 10^{-3}$ for prompt tuning) provides both stable and faster convergence.

Figure 2: Comparison of training loss curves under different learning rate configurations for full fine-tuning and prompt tuning on the SST-2 dataset (Socher et al., 2013) with the base model OPT-1.3b (Zhang et al., 2022). This example illustrates the necessaity of using different learning rates in hybrid-tuning structure.

This example illustrates the practical benefits of considering hybrid smoothness condition and the resulting use of different learning rates in hybrid fine-tuning. It is naturally to ask if this observation can be rigorously supported by the convergence analysis. We address this question in the next subsection.

## 2.3 THEORETICAL ANALYSIS

Recall that our objective is to solve the optimization problem presented in Eq. (1). To handle the generalized smooth structure, we introduce the following definition:

**Definition 2** (Coercive). *A continuous function $f : \mathbb{R}^d \to \mathbb{R}$ is coercive if the sub-level set $\{x \in \mathbb{R}^d \mid f(x) \leq a\}$ is compact for all $a \in \mathbb{R}$.*

In the existing literature of generalized smoothness (Li et al., 2024), this assumption is usually replaced with an equivalent statement: the objective function $f(x, y)$ tends to positive infinity when $(x, y)$ approaches the boundary of its domain. We make the following standard assumptions to regularize the function class and subsequently provide the non-asymptotic convergence analysis.

**Assumption 1** (Regularity Conditions). *The objective function $f(x, y) := \frac{1}{n} \sum_{i=1}^{n} f(x, y; i)$ defined in Eq. (1) satisfies the following conditions:*

*(1) $f(\cdot)$ is coercive.*

*(2) $f(\cdot)$ is bounded below by $f^* := \inf_{(x,y) \in \mathbb{R}^d} f(x, y) > -\infty$.*

*(3) $f(\cdot)$ and each individual loss function $f(\cdot; i)$ are twice continuously differentiable.*

These regularity conditions are essential for several reasons: Coercivity prevents the optimization process from diverging too far. The lower bound guarantees that the optimization problem is well-posed. Twice continuous differentiability allows for the application of various optimization techniques and facilitates theoretical analysis. All of them are standard and widely used in the optimization literature (Li et al., 2024).

**Assumption 2** (Bounded Variance). *There exists $\sigma$ such that for all $x \in \mathbb{R}^d$,*

$$\frac{1}{n} \sum_{i=1}^{n} \|\nabla f(x, y; i) - \nabla f(x, y)\|^2 \leq \sigma^2.$$

This bounded variance assumption is standard in the analysis of reshuffling-type SGD. We note that this assumption could be further weakened to the expected smoothness (Mishchenko et al., 2020; Khaled & Richtárik, 2020). We maintain the current version for the simplicity.

With both assumptions in place, we analyze the complexity of Algorithm 1 under the hybrid smoothness condition (Definition 1). Our main theoretical result is summarized in the following theorem:

**Theorem 1.** *Suppose that Assumption 1 and Assumption 2 hold for the objective function $f(x, y) := \frac{1}{n} \sum_{i=1}^{n} f(x, y; i)$, with satisfying the hybrid smoothness condition (Definition 1). Let $\{(x_t, y_t)\}_{t=1}^{T}$ be the SGD dynamic generated by Algorithm 1 for solving the optimization problem Eq. (1). Let learning rates $\eta_x, \eta_y$ be chosen as*

$$\eta_x \leq \min \left\{ \mathcal{O}(\frac{1}{L_x n d_x}), \mathcal{O}(\frac{1}{\sqrt{T} n L_{x,\max}}) \right\}, \qquad \eta_y \leq \min \left\{ \mathcal{O}(\frac{1}{L_y n}), \mathcal{O}(\frac{1}{\sqrt{T} n L_{y,\max}}) \right\},$$

*and the perturbation stepsize $\mu$ and the smoothness characteristics of the $x$ and $y$ parameters $L_x, L_y, L_{x,\max}, L_{y,\max}$ are specified in the appendix. Let $\delta \in (0, 1)$. If the maximum number of epoch $T$ is chosen as $T \geq \mathcal{O}(\frac{\epsilon^{-2}}{\delta} + \frac{\epsilon^{-4}}{n})$, then with the probability at least $1 - \delta$,*

$$\frac{1}{T} \sum_{t < T} \mathbf{E} \|\nabla f(x_t, y_t)\|^2 \leq \epsilon^2.$$

Given that each epoch processes $n$ data points, the total gradient complexity is $nT \geq \mathcal{O}(\frac{\epsilon^{-2} n}{\delta} + \epsilon^{-4})$. This result is optimal when $\epsilon$ is sufficiently small, aligning with the best-known upper bounds established in previous convergence analyses for both generalized smooth non-convex objectives (Li et al., 2024; Zhang et al., 2019) and $L$-smooth non-convex objectives (Mishchenko et al., 2020; Khaled & Richtárik, 2020). Importantly, it also matches the known lower bound for the SGD algorithm (Arjevani et al., 2023), further confirming its optimality.

**Remark.** On the theoretical side, our analysis highlights the asymmetry between the learning rates $\eta_x$ and $\eta_y$, which arises from the distinct smoothness properties of each variable. This result emphasizes the necessity of adopting tailored learning rate schedules when optimizing modules with hybrid smoothness, an aspect not addressed in standard SGD analysis, validating our empirical observation in Section 2.2. Furthermore, to the best of our knowledge, there is no prior work in the optimization literature that investigates optimization methods under **generalized smoothness** while accounting for **random reshuffling**. Our results constitute the first convergence analysis in this setting.

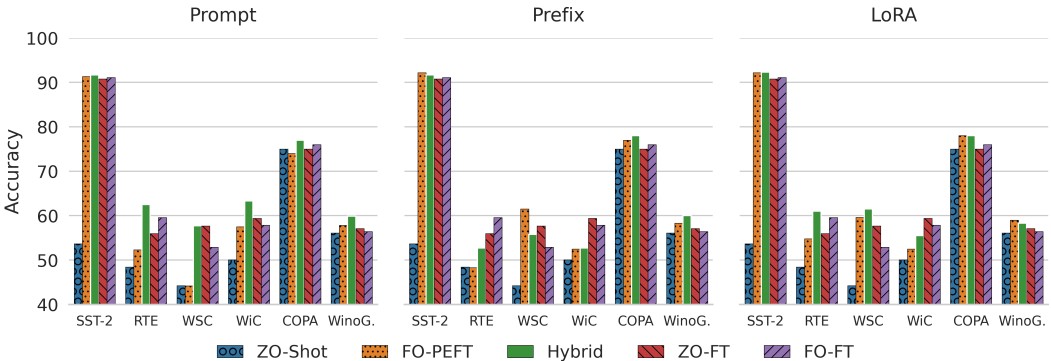

Figure 3: Comparison among Hybrid Fine-Tuning (Hybrid), FO PEFT methods (FO-PEFT), FO full fine-tuning (FO-FT), and ZO full fine-tuning (ZO-FT). In $13/18 \approx 72.2\%$ combinations, Hybrid Fine-Tuning outperforms both ZO and FO full fine-tuning.

## 3 EXPERIMENTS

Following a similar setting of ZO-Bench (Zhang et al., 2024), we evaluate the performance of our proposed method on six representative datasets using three different LLMs. Experimental results show that our proposed method consistently achieves superior performance and faster convergence.

**Tasks & Datasets.** We assessed our approach on 6 representative NLP tasks including the sentiment classification task on the SST2 dataset Socher et al. (2013), the sentence differing task on the WSC dataset Levesque et al. (2012), contextualized word and sense representation and word sense disambiguation task on the WiC dataset Pilehvar & Camacho-Collados (2018), the question answering task on the COPA dataset Roemmele et al. (2011), and the common sense reasoning task on the WinoGrande dataset Sakaguchi et al. (2021).

**Experimental Setting.** Following the methodology of Malladi et al. (2023); Zhang et al. (2024), we assessed our approach on 6 representative NLP tasks including the sentiment classification task on the SST2 dataset Socher et al. (2013), the sentence differing task on the WSC dataset Levesque et al. (2012), contextualized word and sense representation and word sense disambiguation task on the WiC dataset Pilehvar & Camacho-Collados (2018), the question answering task on the COPA dataset Roemmele et al. (2011), and the common sense reasoning task on the WinoGrande dataset Sakaguchi et al. (2021). The models we use in our experiments include OPT-1.3b Zhang et al. (2022), Vicuna-7b Chiang et al. (2023), and LLaMA-7b Zhang et al. (2023b). We compare the performance of our approach against standard PEFT methods including first-order prompt tuning Lester et al. (2021), LoRA tuning Hu et al. (2021), and prefix tuning Li & Liang (2021). For each dataset, we randomly sample 1,000 examples for training, 1,000 examples for evaluation, and 100 examples for development. Performance is evaluated using accuracy or F1 score, as appropriate for each task. All experiments utilize SGD as the optimizer. In the case of prompt tuning and prefix tuning, the prompts are initialized according to the predefined settings in Table E.2 of Malladi et al. (2023), while for LoRA tuning, we initialize with zeros. We perform hyperparameter tuning for all methods and report the best configurations. To ensure a fair comparison, we keep the cardinality of the hyperparameter search spaces identical. We set the maximum number of training steps to 20,000, with early stopping applied when applicable. The detailed hyperparameter setting, overviews of the tasks and PEFT methods, hyper-parameter setting, and the full results are reported in the supplementary materials.

### 3.1 COMPARISON WITH FO AND ZO FULL FINE-TUNING

We begin by evaluating our method on the medium-sized OPT-1.3b model to assess the performance gains over both FO and ZO full-parameter fine-tuning.

Table 1: Experiment results for various fine-tuning methods applied to three large language models across six NLP tasks. Highlighted cells denote the best score for each comparision pair. The left panel (Pairwise Comparison) presents side-by-side comparisons of each Hybrid variant with its corresponding first-order (FO) method (*e.g.* FO-Prompt *vs.* Hybrid-Prompt), enabling direct performance comparisons. The Hybrid method outperforms its FO counterpart in 41 out of 54 cases ($\approx 76\%$). The right panel (First-Order PEFT vs. Hybrid) groups all FO-based methods (Prompt, Prefix, LoRA) separately from their Hybrid counterparts, emphasizing the overall gains from hybrid fine-tuning. In 17 out of 18 comparisons ($\approx 94.5\%$), Hybrid variants yield superior performance.

| Model | Task / Task Type | SST-2 | RTE | WSC | WiC | COPA | WinoG. | Task / Task Type | SST-2 | RTE | WSC | WiC | COPA | WinoG. |
|---|---|---|---|---|---|---|---|---|---|---|---|---|---|---|
| | | | — Classification — | | | — Reasoning — | | | | — Classification — | | | — Reasoning — | |
| **Llama-2-7b** | FO-Prompt | 95.6 | 59.9 | 36.5 | 58.5 | 88.0 | 67.2 | FO-Prompt | 95.6 | 59.9 | 36.5 | 58.5 | 88.0 | 67.2 |
| | Hybrid-Prompt | **95.9** | **59.9** | **61.5** | **64.4** | **88.0** | **68.9** | FO-Prefix | 91.1 | 60.6 | 51.9 | 51.7 | 83.0 | 66.2 |
| | FO-Prefix | 91.1 | 60.6 | **51.9** | **51.7** | 83.0 | **66.2** | FO-LoRA | 94.6 | 62.1 | 60.6 | 61.6 | 84.0 | 68.5 |
| | Hybrid-Prefix | **91.6** | **60.6** | 42.3 | 51.5 | **85.0** | 64.3 | Hybrid-Prompt | **95.9** | 59.9 | 61.5 | 64.4 | 88.0 | 68.9 |
| | FO-LoRA | **94.6** | 62.1 | 60.6 | 61.6 | 84.0 | **68.5** | Hybrid-Prefix | 91.6 | 60.6 | 42.3 | 51.5 | 85.0 | 64.3 |
| | Hybrid-LoRA | 93.4 | **62.5** | **60.6** | **61.7** | **88.0** | 66.3 | Hybrid-LoRA | 93.4 | **62.5** | **60.6** | 61.7 | **88.0** | 66.3 |
| **Vicuna-7b-v1.5** | FO-Prompt | 94.4 | **82.3** | **64.4** | 61.0 | 84.0 | 65.8 | FO-Prompt | 94.4 | 82.3 | 64.4 | 61.0 | 84.0 | 65.8 |
| | Hybrid-Prompt | **95.0** | 70.1 | 55.8 | **64.7** | **84.0** | **66.3** | FO-Prefix | 90.0 | 70.4 | 61.5 | 56.6 | 80.0 | 64.1 |
| | FO-Prefix | 90.0 | 70.4 | 61.5 | **56.6** | 80.0 | 64.1 | FO-LoRA | 94.6 | 80.1 | 53.8 | 58.5 | **85.0** | 66.7 |
| | Hybrid-Prefix | **90.7** | **80.9** | **66.3** | 52.4 | **83.0** | **74.0** | Hybrid-Prompt | **95.0** | 70.1 | 55.8 | 64.7 | 84.0 | 66.3 |
| | FO-LoRA | **94.6** | 80.1 | 53.8 | 58.5 | **85.0** | 66.7 | Hybrid-Prefix | 90.7 | 80.9 | 66.3 | 52.4 | 83.0 | **74.0** |
| | Hybrid-LoRA | 92.2 | **82.0** | **72.1** | **66.8** | 84.0 | **66.7** | Hybrid-LoRA | 92.2 | **82.0** | **72.1** | **66.8** | 84.0 | 66.7 |
| **OPT-1.3b** | FO-Prompt | 91.3 | 52.3 | 44.2 | 57.5 | 74.0 | 57.8 | FO-Prompt | 91.3 | 52.3 | 44.2 | 57.5 | 74.0 | 57.8 |
| | Hybrid-Prompt | **91.7** | **62.5** | **57.7** | **63.3** | **77.0** | **59.9** | FO-Prefix | 92.2 | 48.3 | 61.5 | 52.5 | 77.0 | 58.3 |
| | FO-Prefix | **92.2** | 48.3 | **61.5** | 52.5 | 77.0 | 58.3 | FO-LoRA | 92.2 | 54.8 | 59.6 | 52.5 | 78.0 | 59.0 |
| | Hybrid-Prefix | 91.7 | **52.7** | 55.8 | **52.7** | **78.0** | **60.0** | Hybrid-Prompt | 91.7 | **62.5** | 57.7 | **63.3** | 77.0 | 59.9 |
| | FO-LoRA | 92.2 | 54.8 | 59.6 | 52.5 | 78.0 | **59.0** | Hybrid-Prefix | 91.7 | 52.7 | 55.8 | 52.7 | **78.0** | **60.0** |
| | Hybrid-LoRA | **92.3** | **61.0** | **61.5** | **55.5** | **78.0** | 58.3 | Hybrid-LoRA | **92.3** | 61.0 | **61.5** | 55.5 | **78.0** | 58.3 |

**Results.** We apply our proposed hybrid fine-tuning method to six benchmark tasks using the OPT-1.3b model. As shown in Figure 3, hybrid fine-tuning outperforms its corresponding FO-PEFT counterpart, as well as both FO and ZO full fine-tuning in most scenarios. For example, when using the Prompt Encoder as the PEFT module, hybrid fine-tuning consistently achieves the highest performance across all six tasks, demonstrating robust improvements over all baseline approaches.

## 3.2 PERFORMANCE ON LARGE LANGUAGE MODELS FINE-TUNING

Here we conduct extensive experiments to evaluate the effectiveness of our proposed hybrid fine-tuning approach.

**Results.** In Table 1, we present the comparison between the proposed hybrid fine-tuning and its corresponding FO PEFT fine-tuning. In the aggregate view (right panel), hybrid tuning outperforms FO-based PEFT in 17 out of 18 cases (94.5%), demonstrating its consistent advantage. In the pairwise comparison setting (left panel), where each FO PEFT method is compared with its hybrid variant across three models and six tasks, the hybrid fine-tuning achieves better performance in 41 out of 54 combinations ($\approx 76\%$). These results underscore the effectiveness of hybrid fine-tuning, highlighting its potential as a more robust strategy for adapting LLMs to diverse downstream tasks.

## 3.3 VISUALIZATION OF THE IMPROVED CONVERGENCE RATE

To verify that hybrid fine-tuning converges faster than other methods, we present the training curves (including the training loss, validation accuracy, and the test accuracy) for OPT-1.3B Zhang et al. (2022) model on SST-2 Socher et al. (2013) dataset in Figure 4.

**Results.** We observe that a significant efficiency gain in terms of training steps. For example, as shown in Figure 4, the hybrid fine-tuning takes around 2,500 steps to achieve 90% accuracy, while other methods require at least 12,500 steps to reach the same accuracy. This trend is observed across different tasks, PEFT methods, and model architectures, suggesting that the efficiency of

hybrid tuning scales well (*e.g.* for Vicuna-7b-v1.5 model on the WinoGrande dataset provided in the supplementary materials).

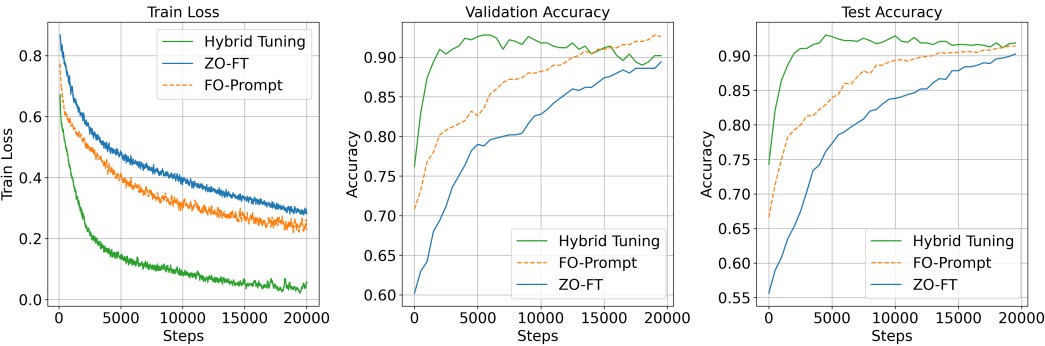

Figure 4: Training curves for OPT-1.3B model with the prompt tuning on the SST2 dataset. The Hybrid Fine-Tuning method achieves significantly faster convergence than the other two baselines.

Table 2: Comparison of theoretical, asymptotical, and empirical memory usage for different fine-tuning optimizers. $|x|$ and $|y|$ denote the parameter counts of the base LLM and the PEFT module, respectively, with $|y|/|x| \to 0$. $|a|$ and $|b|$ are the per-layer gradient states kept during optimization.

| Optimizer | Theoretical Memory | Asymptotical Memory | Empirical Memory | Consumed GPU |
|---|---|---|---|---|
| FO-SGD (LLM) | $\sum_\ell \max\{|a_\ell|, |x_\ell|\} + |x|$ | $\sum_\ell \max\{|a_\ell|, |x_\ell|\}$ | 54 GB | $2 \times$A6000 |
| ZO-SGD (LLM) | $\max_\ell |x_\ell|$ | $\max_\ell |x_\ell|$ | 32 GB | $1 \times$A6000 |
| FO-SGD (Prompt) | $\sum_\ell \max\{|b_\ell|, |y_\ell|\} + |x|$ | $|x|$ | 46 GB | $1 \times$A6000 |
| Hybrid ZO-SGD (LLM+Prompt) | $\sum_\ell \max\{|b_\ell|, |y_\ell|\} + \max_\ell |x_\ell| + |x|$ | $|x|$ | 46 GB | $1 \times$A6000 |

## 3.4 MEMORY USAGE ANALYSIS

In this section, we consider the memory usage of the hybrid fine-tuning approach. Let $|x_\ell|$ and $|y_\ell|$ denote the parameter sizes of the $\ell$-th layer in the base LLM and the PEFT module, respectively, with $|x| := \sum_\ell |x_\ell| \gg |y| := \sum_\ell |y_\ell|$ in most practical scenarios. During first-order optimization, each computational graph node stores local gradient states, represented as $a_\ell$ for the LLM and $b_\ell$ for the PEFT module. A key observation is that despite updating additional parameters with the inclusion of both the base LLM and the PEFT module, the hybrid fine-tuning approach does not increase the asymptotical memory usage (i.e. as $\frac{|y|}{|x|} \to 0$). While the theoretical memory footprint of Hybrid ZO-SGD (LLM+PEFT) is $|x| + |y|$, it remains dominated by $|x|$ in practice. Thus, the hybrid fine-tuning method enables updating more parameters without significantly increasing memory consumption, ensuring scalability even for large-scale LLMs.

**Results.** Furthermore, our empirical results confirm this observation. Table 2 reports both the theoretical memory requirements of several fine-tuning strategies and their actual peak GPU memory usage when fine-tuning Llama-2-7B on the SST-2 dataset. The hybrid approach not only significantly reduces memory overhead compared to FO full fine-tuning ($\approx 15\%$ reduction), but also matches the memory footprint of FO prompt tuning (Lester et al., 2021).

## 3.5 EXTENDED COMPARISON OF GRADIENT LIPSCHITZ CONSTANT

In this subsection, we present extended experiments to further examine the local geometry of the optimization landscape. Specifically, we directly estimate and compare the gradient Lipschitz constants

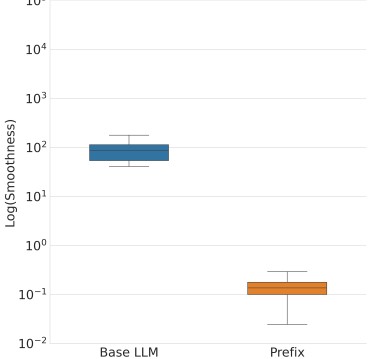 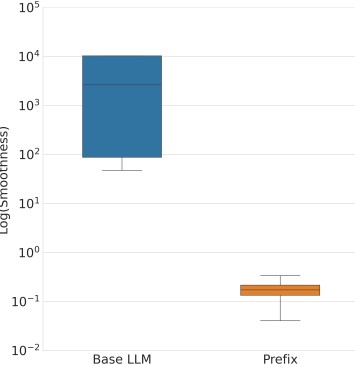

Figure 5: Extended comparison of gradient Lipschitz constant $L$ for OPT-1.3b (Left) and LLaMa-2-7b (Right) in different modules (Base LLM and Prefix Tuning). The base LLM exhibits a significantly larger Lipschitz constant, further confirming our observation in Figure 1b.

Table 3: Pairwise comparison between FO and Hybrid variants on OPT-1.3b across six NLP tasks. Additionally compared to Table 1, we include the Adam optimizer as the baseline. Notably, our proposed method still achieves advanced performance without adopting the Adam optimizer. In principle, our approach can be further enhanced by replacing the SGD update for the PEFT module with Adam to further accelerate the training.

Table 4: Empirical memory usage of different optimizers for the OPT-1.3b model. We note that the Adam optimizer takes three-time memory of the SGD optimizer due to the need to store two additional state variables (the first and second moments) for each model parameter.

| Model | Task | SST-2 | RTE | WSC | WiC | COPA | WinoG. |
|---|---|---|---|---|---|---|---|
| OPT-1.3b | FO-LoRA (Adam) | 91.7 | 58.6 | 58.7 | **64.1** | 66.0 | **60.1** |
| | FO-LoRA (SGD) | 92.2 | 54.8 | 59.6 | 52.5 | 78.0 | 59.0 |
| | Hybrid-LoRA (SGD) | **92.3** | **61.0** | **61.5** | 55.5 | **78.0** | 58.3 |

| Optimizer | Emp. Mem. |
|---|---|
| FO-SGD (LLM) | 11.2 GB |
| ZO-SGD (LLM) | 6.8 GB |
| FO-Adam (LoRA) | 11.0 GB |
| Hybrid ZO-SGD (LLM+LoRA) | 10.7 GB |

associated with the base model parameters ($x$) and the PEFT parameters ($y$) variables across multiple models (OPT-1.3b and LLaMa-2-7b). As shown in Figure 5, these additional results consistently support our hybrid smoothness assumption by showing that the $x$-coordinates exhibit noticeably larger Lipschitz constants, indicating the necessity of applying a different learning rate scale.

## 3.6 EXTENDED COMPARISON WITH THE ADAM+LORA BASELINE

In our previous comparison, using SGD across all methods was driven by our theoretical focus; the core of our contribution is the convergence analysis for our hybrid method under the novel Hybrid Smoothness condition , which we developed specifically for SGD with Random Reshuffling.

To further validate the effectiveness of our approach, we conduct an additional set of experiments comparing Hybrid-LoRA with the Adam+LoRA baseline. This extended evaluation examines whether the performance gains observed in our main results persist under the widely applied optimization settings. The results reported in Table 3 demonstrate that Hybrid-LoRA still maintains its advantage while using smaller memory (as indicated by Table 4). We also emphasize that our framework is extensible, and one could construct a "Hybrid-Adam" variant by replacing the SGD update for the PEFT module with Adam. We view this as an interesting direction for future work.

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

## A    RELATED WORK

**Zeroth-Order Optimization in Fine-Tuning LLMs**    Recent work has explored ZO optimization methods for fine-tuning LLMs, which aligns with our approach of using ZO methods for the LLM component in hybrid fine-tuning. Malladi et al. (2023) demonstrated the compatibility of zeroth-order methods with both full fine-tuning and PEFTs. This laid the groundwork for our hybrid approach that combines zeroth-order LLM updates with first-order PEFT updates. Zhang et al. (2024) provided a comprehensive benchmark for ZO optimization in LLM fine-tuning, offering valuable insights that informed our experimental design. Ling et al. (2024) combines the ZO fine-tuning of LLMs with the federated learning. Several studies have incorporated variance reduction techniques (Gautam et al., 2024) into ZO methods or second-order method (Zhao et al., 2024) to enhance stability and convergence in fine-tuning LLMs. While we focus on a different aspect, these stability improvements could easily be integrated into our hybrid framework. Existing literature (Liu et al., 2024; Guo et al., 2024; Zhang et al., 2024) also discusses the sparsity of pre-trained LLMs, which further enhances the performance of ZO optimization approach.

**Generalized Smoothness of Large Machine Learning Models**    The concept of generalized smoothness has emerged as a crucial theoretical framework for understanding the optimization landscape of large machine learning models, including LLMs. Recent studies have shown that traditional smoothness assumptions often fail to capture the complex optimization landscape of deep neural networks (Zhang et al., 2019; Li et al., 2024). More explicitly, Zhang et al. (2019) demonstrated that the local smoothness constant in neural networks is often proportional to the gradient norm, challenging the conventional assumption of uniform smoothness. This insight aligns with our observations in hybrid fine-tuning, where different components of the model (LLM and PEFT modules) exhibit distinct smoothness properties. Li et al. (2024) introduced a generalized smoothness condition that allows for non-uniform smoothness across the parameter space, which is more representative of the behavior observed in practice for large models. This work provides a foundation for our hybrid generalized smoothness framework, which extends these ideas to account for the heterogeneous nature of joint LLM and PEFT optimization.

## B    NOTATIONS

In this paper, the optimization problem is formulated as minimizing $f(x, y)$, where $x \in \mathbb{R}^{d_x}$ represents the parameters of the base language model and $y \in \mathbb{R}^{d_y}$ represents the parameters of the PEFT module. The function $f$ is assumed to have hybrid generalized smoothness, characterized by non-negative, non-decreasing sub-quadratic functions $\ell_x$ and $\ell_y$ (Definition 1). In the SGD, we consider epoch-wise optimization algorithm described in Algorithm 1. This approach ensures us to access each data point exactly once over an entire epoch, which is particularly common is the data loader provided by existing modern machine learning frameworks such as PyTorch and TensorFlow. Here, $\eta_x$ and $\eta_y$ denote the learning rates for $x$ and $y$ respectively, $T$ is the number of epochs, and $n$ is the dataset size. We $\hat{\nabla}_x f$ to denote the zeroth-order gradient estimator for $x$, while $\nabla_y f$ represents the standard gradient for $y$. With these given, for each epoch $t$, we define the following notations:

$$g_t = \sum_{i=1}^{n} \nabla_x f(x_{t,i}, y_{t,i}; \xi_{t,i}), \quad \hat{g}_t = \sum_{i=1}^{n} \hat{\nabla}_x f(x_{t,i}, y_{t,i}; \xi_{t,i}),$$

$$h_t = \sum_{i=1}^{n} \nabla_y f(x_{t,i}, y_{t,i}; \xi_{t,i}).$$

Here, $g_t$ represents the true gradient with respect to $x$ accumulated over an entire epoch. It captures the overall direction of stochastic gradient descent for the $x$ parameters across all samples in the epoch. $\hat{g}_t$ is an estimate of this gradient. In practice, we often don't have access to the true gradient and must rely on estimates. The difference between $g_t$ and $\hat{g}_t$ quantifies the estimation error in our gradient calculations. $h_t$ is the true gradient with respect to $y$ accumulated over the epoch.

## C  SUPPORTING LEMMAS

In this section, we present several lemmas used to build our convergence analysis. Lemma 1, Lemma 2, Lemma 3, and Lemma 4 are fundamental properties of generalized smoothness provided by Li et al. (2024). We adapt them to the setting of hybrid system fine-tuning.

**Lemma 1** (The generalized version of Lemma 3.3 from Li et al. (2024))**.** *Let $f : \mathbb{R}^d = \mathbb{R}^{d_x} \times \mathbb{R}^{d_y} \to \mathbb{R}$ be a twice continuously differentiable function satisfying the hybrid generalized smoothness properties. Suppose that $(x, y) \in \mathbb{R}^d$ satisfies $\|\nabla f(x, y)\| \leq G$. Then there exist non-negative constant $L_x = \ell_x(G)$ and $L_y = \ell_y(G)$ such that for all $(x_1, y_1), (x_2, y_2) \in \mathcal{B}(x, \frac{G}{L_x}) \times \mathcal{B}(y, \frac{G}{L_y})$:*

*1. $\|\nabla_x f(x_1, y') - \nabla_x f(x_2, y')\| \leq L_x \|x_1 - x_2\|$, for all $y' \in \mathbb{R}^{d_y}$.*

*2. $\|\nabla_y f(x', y_1) - \nabla_y f(x', y_2)\| \leq L_y \|y_1 - y_2\|$, for all $x' \in \mathbb{R}^{d_x}$.*

*3. Let $I_d$ represent the identity matrix with the size $d \times d$.*

$$f(x_1, y_1) \leq f(x_2, y_2) + \left\langle \nabla f(x_2, y_2), \begin{bmatrix} x_1 - x_2 \\ y_1 - y_2 \end{bmatrix} \right\rangle$$
$$+ \frac{1}{2} [x_1 - x_2 \quad y_1 - y_2] \begin{bmatrix} L_x I_{d_x} & 0 \\ 0 & L_y I_{d_y} \end{bmatrix} \begin{bmatrix} x_1 - x_2 \\ y_1 - y_2 \end{bmatrix}.$$

*Proof.* Let $(x, y) \in \mathbb{R}^d = \mathbb{R}^{d_x} \times \mathbb{R}^{d_y}$ be arbitrary. By the assumption of twice continuous differentiability and the mean value theorem, we have

$$\nabla_x f(x_2, y) - \nabla_x f(x_1, y) = \int_0^1 \nabla_{xx}^2 f(x_1 + t(x_2 - x_1), y)(x_2 - x_1) dt.$$

Taking the norm of both sides and applying the generalized smoothness of $f$ (Definition 1), we obtain

$$\|\nabla_{xx}^2 f(x, y)\| \leq \ell_x(\|\nabla f(x, y)\|) \leq L_x,$$

where the last inequality is by the monotonicity of $\ell_x$ and the bounded gradient condition. We apply this inequality to the integral yields the first inequality. The second inequality for the y-gradient is obtained similarly. For the third inequality, we still consider the mean value theorem:

$$f(x_1, y_1) - f(x_2, y_2) = \int_0^1 \left\langle \nabla f(z(t)), \begin{bmatrix} x_1 - x_2 \\ y_1 - y_2 \end{bmatrix} \right\rangle dt$$
$$= \int_0^1 \left[ \left\langle \nabla f(x_2, y_2), \begin{bmatrix} x_1 - x_2 \\ y_1 - y_2 \end{bmatrix} \right\rangle + \left\langle \nabla f(z(t)) - \nabla f(x_2, y_2), \begin{bmatrix} x_1 - x_2 \\ y_1 - y_2 \end{bmatrix} \right\rangle \right] dt$$
$$= \left\langle \nabla f(x_2, y_2), \begin{bmatrix} x_1 - x_2 \\ y_1 - y_2 \end{bmatrix} \right\rangle + \int_0^1 \left\langle \nabla f(z(t)) - \nabla f(x_2, y_2), \begin{bmatrix} x_1 - x_2 \\ y_1 - y_2 \end{bmatrix} \right\rangle dt$$
$$\leq \left\langle \nabla f(x_2, y_2), \begin{bmatrix} x_1 - x_2 \\ y_1 - y_2 \end{bmatrix} \right\rangle + L_y \|y_1 - y_2\|^2 \int t dt + L_x \|x_1 - x_2\|^2 \int t dt,$$

where $z(t) := (1 - t) \begin{bmatrix} x_2 \\ y_2 \end{bmatrix} + t \begin{bmatrix} x_1 \\ y_1 \end{bmatrix}$ for $0 \leq t \leq 1$. Then the proof is completed by re-arranging this inequality. $\square$

**Lemma 2** (The generalized version of Lemma 3.5 from Li et al. (2024))**.** *Let $f : \mathbb{R}^{d_x} \times \mathbb{R}^{d_y} \to \mathbb{R}$ be a twice continuously differentiable function satisfying the hybrid generalized smoothness properties. Let $f^* = \inf_{x,y} f(x, y)$ be the global minimum of $f$. Then, for all $(x, y) \in \mathbb{R}^{d_x} \times \mathbb{R}^{d_y}$, the following inequalities hold:*

*1. $\|\nabla_x f(x, y)\|^2 \leq 2\ell_x(2\|\nabla f(x, y)\|) \cdot (f(x, y) - f^*)$*

*2. $\|\nabla_y f(x, y)\|^2 \leq 2\ell_y(2\|\nabla f(x, y)\|) \cdot (f(x, y) - f^*)$*

*3. $\frac{1}{2} [\nabla f(x, y)]^\top \begin{bmatrix} \frac{I_{d_x}}{\ell_x(2\|\nabla f(x,y)\|)} & 0 \\ 0 & \frac{I_{d_y}}{\ell_y(2\|\nabla f(x,y)\|)} \end{bmatrix} \nabla f(x, y) \leq f(x, y) - f^*.$*

*Proof.* The first and the second inequalities are directly implied by Lemma 3.5 from Li et al. (2024) by projecting the objective function $f$ to a subspace of the domain. Here, we provide the proof for the third inequality. By Lemma 1 where we choose $G = \|\nabla f(x, y)\|$, we have that for any $(x_1, y_1), (x_2, y_2) \in \mathcal{B}(x, \frac{G}{L_x}) \times \mathcal{B}(y, \frac{G}{L_y})$,

$$f(x_1, y_1) \leq f(x_2, y_2) + \left\langle \nabla f(x_2, y_2), \begin{bmatrix} x_1 - x_2 \\ y_1 - y_2 \end{bmatrix} \right\rangle + \frac{1}{2} \begin{bmatrix} x_1 - x_2 & y_1 - y_2 \end{bmatrix} \begin{bmatrix} L_x I_{d_x} & 0 \\ 0 & L_y I_{d_y} \end{bmatrix} \begin{bmatrix} x_1 - x_2 \\ y_1 - y_2 \end{bmatrix}.$$

Choosing $(x_2, y_2) = (x, y)$, $x_1 = x - \frac{\nabla_x f(x,y)}{\ell_x(2\|\nabla f(x,y)\|)}$, and $y_1 = y - \frac{\nabla_y f(x,y)}{\ell_y(2\|\nabla f(x,y)\|)}$, we obtain

$$f^* \leq f(x - \frac{\nabla_x f(x,y)}{\ell_x(2\|\nabla f(x,y)\|)}, y - \frac{\nabla_y f(x,y)}{\ell_y(2\|\nabla f(x,y)\|)})$$

$$\leq f(x, y) - \frac{1}{2}[\nabla f(x,y)]^\top \begin{bmatrix} \frac{I_{d_x}}{\ell_x(2\|\nabla f(x,y)\|)} & 0 \\ 0 & \frac{I_{d_y}}{\ell_y(2\|\nabla f(x,y)\|)} \end{bmatrix} \nabla f(x, y).$$

Then the proof is completed. $\qquad\square$

**Lemma 3** (The generalized version of Corollary 3.6 from Li et al. (2024)). *Let $f : \mathbb{R}^{d_x} \times \mathbb{R}^{d_y} \to \mathbb{R}$ be a twice continuously differentiable function satisfying the hybrid generalized smoothness properties. Suppose that $f(x, y) - f^* \leq F$ for some $(x, y) \in \mathbb{R}^d$ and $F \geq 0$. Denoting $G := \sup\{u \geq 0 \mid u^2 \leq 2\max(\ell_x, \ell_y)(u) \cdot F\}$, then $\|\nabla f(x, y)\| \leq G < \infty$.*

*Proof.* Let $\max(\ell_x, \ell_y)(u) := \max\{\ell_x(u), \ell_y(u)\}$. Since both $\ell_x$ and $\ell_y$ are sub-quadratic, it concludes $G$ is finite (by Corollary 3.6 from Li et al. (2024)). From Lemma 2, we have

$$\frac{1}{2}[\nabla f(x,y)]^\top \begin{bmatrix} \frac{I_{d_x}}{\max(\ell_x, \ell_y)(2\|\nabla f(x,y)\|)} & 0 \\ 0 & \frac{I_{d_y}}{\max(\ell_x, \ell_y)(2\|\nabla f(x,y)\|)} \end{bmatrix} \nabla f(x, y)$$

$$\leq \frac{1}{2}[\nabla f(x,y)]^\top \begin{bmatrix} \frac{I_{d_x}}{\ell_x(2\|\nabla f(x,y)\|)} & 0 \\ 0 & \frac{I_{d_y}}{\ell_y(2\|\nabla f(x,y)\|)} \end{bmatrix} \nabla f(x, y)$$

$$\leq f(x, y) - f^*.$$

Therefore, we obtain

$$\|\nabla f(x, y)\|^2 \leq 2\max(\ell_x, \ell_y)(2\nabla f(x, y)) \cdot F.$$

It concludes that if the function value is bounded, then the gradient is also bounded. $\qquad\square$

Here, we summarize the previous results in the following lemma. The constant $G$ (determined by the function value upper bound $F$) is defined in Lemma 3 and the constant $L_x$ and $L_y$ (determined by the gradient norm upper bound $G$) is defined in Lemma 1.

**Lemma 4.** *Suppose that Assumption 1 holds for the objective function $f(x, y) := \frac{1}{n}\sum_{i=1}^n f(x, y; i)$, with all individual loss functions $f(\cdot; i)$ are twice continuously differentiable and satisfy the hybrid generalized smoothness properties. Let $\mathcal{G}_F := \{(x, y) \in \mathbb{R}^d \mid f(x, y) - f^* \leq F\}$. Then the following statements hold:*

1. *The objective function $f(\cdot)$ has $G$-bounded gradient over $\mathcal{G}_F$; that is, $\|\nabla f(x, y)\| \leq G$ for all $(x, y) \in \mathcal{G}_F$.*

2. *The objective function $f(\cdot)$ has $(L_x, L_y)$-Lipschitz gradient over $\mathcal{G}_F$; that is, $\|\nabla_x f(x, y) - \nabla_x f(x', y)\| \leq L_x \|x - x'\|$ and $\|\nabla_y f(x, y) - \nabla_y f(x, y')\| \leq L_y \|y - y'\|$ for all $(x, y), (x', y') \in \mathcal{G}_F$.*

3. *The individual loss function $f(\cdot; i)$ has $(G_{x,\max}, G_{y,\max})$-bounded gradient over $\mathcal{G}_F$; that is, $\|\nabla_x f(x, y; \xi)\| \leq G_{x,\max}$ and $\|\nabla_y f(x, y; \xi)\| \leq G_{y,\max}$ for all $(x, y) \in \mathcal{G}_F$ and all $\xi \in \{1, 2, \ldots, n\}$.*

4. *The individual loss function $f(\cdot; i)$ has $(L_{x,\max}, L_{y,\max})$-Lipschitz gradient over $\mathcal{G}_F$; that is, $\|\nabla_x f(x, y; \xi) - \nabla_x f(x', y; \xi)\| \leq L_{x,\max} \|x - x'\|$ and $\|\nabla_y f(x, y; \xi) - \nabla_y f(x, y'; \xi)\| \leq L_{y,\max} \|y - y'\|$ for all $(x, y) \in \mathcal{G}_F$ and all $\xi \in \{1, 2, \ldots, n\}$.*

*Proof.* By Assumption 1, $\mathcal{G}_F$ is a compact set. By the twice continuous differentiability of the objective function $f(\cdot)$ (and all individual loss functions $f(\cdot; i)$), all statements holds by its continuity. More precise evaluation is given in Lemma 1 for $L_x$ and $L_y$, and in Lemma 3 for $G$. $\qquad\square$

The following lemma characterizes the accuracy of zeroth-order gradient estimation. We note that the choice of zeroth-order gradient estimator is not the crucial part in our analysis; the following gradient estimation method can be replaced with any common zeroth-order optimization techniques, including the mini-batch zeroth-order gradient estimation Nesterov & Spokoiny (2017), the uniform smoothing Gasnikov et al. (2022), and the variance reduction Liu et al. (2018).

**Lemma 5.** *Let $f : \mathbb{R}^d \to \mathbb{R}$ be a function with twice continuous differentiability. Define the two-point zeroth-order gradient estimator of $\nabla f(x)$ as*

$$\hat{\nabla} f(x) := \frac{1}{\mu} \left[ f(x + \mu v) - f(x) \right] v,$$

*where $\mu > 0$ is the perturbation stepsize, $v \in \mathbb{R}^d$ is a Gaussian vector with the covariance matrix $I_d$. Suppose that $f$ has $G$-bounded gradient and $L$-Lipschitz gradient at $x$. Then*

1. $\mathbf{E}\langle g, \hat{\nabla} f(x) - \nabla f(x) \rangle \leq \frac{\mu}{2} L(d+3)^{3/2} \|g\|$, *for any $g \in \mathbb{R}^d$.*

2. $\mathbf{E}\|\hat{\nabla} f(x) - \nabla f(x)\|^2 \leq 32d\|\nabla f(x)\|^2 + 108\mu^2 L^2 d^4$.

*Proof.* Throughout this proof, we follow the *random gradient-free oracles* given by Nesterov & Spokoiny (2017). That is, define

$$f_\mu(x) = \mathbf{E}_{v \sim N(0, I_d)} f(x + \mu v);$$

then the gradient estimator $\hat{\nabla} f(x)$ is an unbiased estimator of $\nabla f_\mu(x)$. For the first inequality, we have

$$\mathbf{E}\langle g, \hat{\nabla} f(x) - \nabla f(x) \rangle \overset{(i)}{=} \mathbf{E}\langle g, \nabla f_\mu(x) - \nabla f(x) \rangle$$
$$\overset{(ii)}{=} \frac{\mu}{2} L(d+3)^{3/2} \|g\|.$$

where (i) applies the unbiasedness of Gaussian smoothing and (ii) applies Lemma 3 from Nesterov & Spokoiny (2017). For the second inequality, we have

$$\mathbf{E}\|\hat{\nabla} f(x) - \nabla f(x)\|^2 \leq 2\mathbf{E}\|\hat{\nabla} f(x)\|^2 + 2\|\nabla f(x)\|^2$$
$$\overset{(i)}{\leq} 8(d+4)\|\nabla f_\mu(x)\|^2 + 6\mu^2 L^2 (d+4)^3 + 2\|\nabla f(x)\|^2$$
$$\overset{(ii)}{\leq} 32d\|\nabla f(x)\|^2 + 108\mu^2 L^2 d^4.$$

where (i) applies Lemma 5 from Nesterov & Spokoiny (2017) and (ii) again applies Lemma 3 from Nesterov & Spokoiny (2017). $\qquad\square$

**Lemma 6.** *Suppose that Assumption 1 and Assumption 2 hold for the objective function $f(x, y) := \frac{1}{n} \sum_{i=1}^n f(x, y; i)$, with all individual loss functions $f(\cdot; i)$ are twice continuously differentiable and satisfy the hybrid generalized smoothness properties. Let*

$$\epsilon_t = \frac{1}{n} \sum_{i=1}^n \hat{\nabla} f(x_{t,i}, y_{t,i}; \xi_{t,i}) - \frac{1}{n} \sum_{i=1}^n \nabla f(x_{t,i}, y_{t,i}; \xi_{t,i}) + \frac{1}{n} \sum_{i=1}^n \nabla f(x_{t,i}, y_{t,i}; \xi_{t,i}) - \nabla f(x_t, y_t),$$

*be the gradient approximation error over the $t$-th epoch. Given any $F, H > 0$, define the stopping time as $\tau = \tau_1 \wedge \tau_2$, where $\tau_1 := \min_t \{ t \mid f(x_{t+1}, y_{t+1}) - f^* > F \} \wedge T$ and $\tau_2 := \min_t \{ t \mid \|\epsilon_t\| > H \} \wedge T$. Let the learning rates satisfy $\eta_x \leq \min\{\frac{1}{2L_{x,\max} n}, \frac{1}{384 L_x n d_x}\}$ and $\eta_y \leq \frac{1}{2L_{y,\max} n}$ and the perturbation stepsize $\mu \leq \frac{G}{L_x} \frac{6}{d_x^{3/2}}$. Then*

$$f(x_\tau, y_\tau) - f^* + \sum_{t<\tau} [\nabla f(x_t, y_t)]^\top \begin{bmatrix} \frac{n}{4} \eta_x I_{d_x} & 0 \\ 0 & \frac{n}{3} \eta_y I_{d_y} \end{bmatrix} \nabla f(x_t, y_t)$$

$$\leq f_0 - f^* + \left[ \frac{\sigma^2}{2} n^2 \left[ \eta_y^3 L_{y,\max}^2 + \eta_x^3 L_{x,\max}^2 \right] + o(\mu) \right] T,$$

*where $o(\mu) \leq 3\eta_x \mu n L_x d_x G$ is a small error term when $\mu$ is chosen small.*

*Proof.* For arbitrary stopping time $\tau$, we start from the smoothness given by Lemma 1:

$$f(x_{t+1}, y_{t+1}) - f(x_t, y_t)$$

$$\leq \left\langle \nabla f(x_t, y_t), \begin{bmatrix} x_{t+1} - x_t \\ y_{t+1} - y_t \end{bmatrix} \right\rangle + \frac{1}{2} \begin{bmatrix} x_{t+1} - x_t & y_{t+1} - y_t \end{bmatrix} \begin{bmatrix} L_x I_{d_x} & 0 \\ 0 & L_y I_{d_y} \end{bmatrix} \begin{bmatrix} x_{t+1} - x_t \\ y_{t+1} - y_t \end{bmatrix}$$

$$= \langle \nabla_x f(x_t, y_t), x_{t+1} - x_t \rangle + \langle \nabla_y f(x_t, y_t), x_{t+1} - x_t \rangle + \frac{L_x}{2} \|x_{t+1} - x_t\|^2 + \frac{L_y}{2} \|y_{t+1} - y_t\|^2$$

$$\overset{(i)}{=} -\eta_x n \langle \nabla_x f(x_t, y_t), \frac{\hat{g}_t}{n} - \frac{g_t}{n} \rangle - \eta_x n \langle \nabla_x f(x_t, y_t), \frac{g_t}{n} \rangle + \eta_x^2 L_x n^2 \|\frac{\hat{g}_t}{n} - \frac{g_t}{n}\|^2 + \eta_x^2 L_x n^2 \|\frac{g_t}{n}\|^2$$

$$- \eta_y n \langle \nabla_y f(x_t, y_t), \frac{h_t}{n} \rangle + \eta_y^2 L_y n^2 \|\frac{h_t}{n}\|^2,$$

where (i) we applies the derivation of Eq.(38) from Mishchenko et al. (2020) with setting $\eta_x \leq \frac{1}{2L_x}$ and $\eta_y \leq \frac{1}{2L_y}$. We note that the y parameter update doesn't involve the gradient estimation; so, we keep the original stochastic gradient $h_t$ for this step. Let $\mathcal{E}_1 = -\eta_x n \langle \nabla_x f(x_t, y_t), \frac{\hat{g}_t}{n} - \frac{g_t}{n} \rangle$ and $\mathcal{E}_2 = \eta_x^2 L_x n^2 \|\frac{\hat{g}_t}{n} - \frac{g_t}{n}\|^2$, representing the errors caused by the zeroth-order gradient estimation. Then we obtain

$$f(x_{t+1}, y_{t+1}) - f(x_t, y_t) \leq -\eta_x n \langle \nabla_x f(x_t, y_t), \frac{g_t}{n} \rangle + \eta^2 L_x n^2 \|\frac{g_t}{n}\|^2 + \mathcal{E}_1 + \mathcal{E}_2$$

$$- \eta_y n \langle \nabla_x f(x_t, y_t), \frac{h_t}{n} \rangle + \eta^2 L_y n^2 \|\frac{h_t}{n}\|^2.$$

Then we set $\eta_x \leq \frac{1}{2L_x n}$ and $\eta_y \leq \frac{1}{2L_y n}$. By Eq.(39) from Mishchenko et al. (2020),

$$f(x_{t+1}, y_{t+1}) - f(x_t, y_t) + \frac{\eta_x n}{2} \|\nabla_x f(x_t, y_t)\|^2 + \frac{\eta_y n}{2} \|\nabla_y f(x_t, y_t)\|^2$$

$$\leq \frac{\eta_x n}{2} \left\| \frac{g_t}{n} - \nabla_x f(x_t, y_t) \right\|^2 + \frac{\eta_y n}{2} \left\| \frac{h_t}{n} - \nabla_y f(x_t, y_t) \right\|^2 + \mathcal{E}_1 + \mathcal{E}_2.$$

Then we take expectation on both sides and decompose $\left\| \nabla_x f(x_t, y_t) - \frac{g_t}{n} \right\|^2$ using Lemma 1 with the Lipschitz constant $L_{x,\max}$ and $\left\| \nabla_y f(x_t, y_t) - \frac{g_t}{n} \right\|^2$ with the Lipschitz constant $L_{y,\max}$; more explicitly, we have

$$\left\| \nabla_x f(x_t, y_t) - \frac{g_t}{n} \right\|^2 = \left\| \frac{1}{n} \sum_{i=1}^n \nabla_x f(x_{t,0}, y_{t,0}; \xi_{t,i}) - \frac{1}{n} \sum_{i=1}^n \nabla_x f(x_{t,i}, y_{t,i}; \xi_{t,i}) \right\|^2$$

$$\leq \frac{1}{n} \sum_{i=1}^n \|\nabla_x f(x_{t,0}, y_{t,0}; \xi_{t,i}) - \nabla_x f(x_{t,i}, y_{t,i}; \xi_{t,i})\|^2$$

$$\leq \frac{L_{x,\max}^2}{n} \sum_{i=1}^n \|x_{t,0} - x_{t,i}\|^2.$$

Applying Assumption 2 and Lemma 5 from Mishchenko et al. (2020) to bound $\frac{L_{x,\max}^2}{n} \sum_{i=1}^n \mathbf{E} \|x_{t,0} - x_{t,i}\|^2$, we obtain

$$f(x_{t+1}, y_{t+1}) - f(x_t, y_t) + \frac{\eta_x n}{2} \|\nabla_x f(x_t, y_t)\|^2 + \frac{\eta_y n}{2} \|\nabla_y f(x_t, y_t)\|^2$$

$$\leq \frac{\eta_x n}{2} \frac{L_{x,\max}^2}{n} [\eta_x^2 n^3 \|\nabla_x f(x_t, y_t)\|^2 + \eta_x^2 n^2 \sigma^2] + \frac{\eta_y n}{2} \frac{L_{y,\max}^2}{n} [\eta_y^2 n^3 \|\nabla f(x_t, y_t)\|^2 + \eta_y^2 n^2 \sigma^2] + \mathbf{E}\mathcal{E}_1 + \mathbf{E}\mathcal{E}_2.$$

We re-write this inequality into the matrix form.

$$f(x_{t+1}, y_{t+1}) - f(x_t, y_t) + [\nabla f(x_t, y_t)]^\top \begin{bmatrix} \frac{\eta_x n}{2} & 0 \\ 0 & \frac{\eta_y n}{2} \end{bmatrix} \nabla f(x_t, y_t)$$

$$\leq \frac{\sigma^2}{2} n^2 [\eta_x^3 L_{x,\max}^2 + \eta_y^3 L_{y,\max}^2] + \mathbf{E}\mathcal{E}_1 + \mathbf{E}\mathcal{E}_2 + [\nabla f(x_t, y_t)]^\top \begin{bmatrix} \frac{\eta_x^3 n^3 L_{x,\max}^2}{2} I_{d_x} & 0 \\ 0 & \frac{\eta_y^3 n^3 L_{y,\max}^2}{2} I_{d_y} \end{bmatrix} \nabla f(x_t, y_t).$$

When choosing $\eta_x \leq \frac{1}{2L_{x,\max}n}$ and $\eta_y \leq \frac{1}{2L_{y,\max}n}$, it ensures that

$$\frac{n}{3}\begin{bmatrix} \eta_x I_{d_x} & 0 \\ 0 & \eta_y I_{d_x} \end{bmatrix} \preceq \begin{bmatrix} \frac{\eta_x n}{2} I_{d_x} & 0 \\ 0 & \frac{\eta_y n}{2} I_{d_y} \end{bmatrix} - \begin{bmatrix} \frac{\eta_x^3 n^3 L_{x,\max}^2}{2} I_{d_x} & 0 \\ 0 & \frac{\eta_y^3 n^3 L_{y,\max}^2}{2} I_{d_y} \end{bmatrix}.$$

Therefore, we let $\Lambda^2 = \frac{n}{3}\begin{bmatrix} \eta_x I_{d_x} & 0 \\ 0 & \eta_y I_{d_y} \end{bmatrix}$ be a PSD matrix. Then we obtain

$$f(x_{t+1}, y_{t+1}) - f(x_t, y_t) + \|\Lambda \nabla f(x_t, y_t)\|^2 \leq \frac{\sigma^2}{2}n^2 \left[\eta_x^3 L_{x,\max}^2 + \eta_y^3 L_{y,\max}^2\right] + \mathbf{E}\mathcal{E}_1 + \mathbf{E}\mathcal{E}_2.$$

Then we apply Lemma 5 to bound $\mathbf{E}\mathcal{E}_1$ and $\mathbf{E}\mathcal{E}_2$, respectively. By the stopping time construction, we have $\|\nabla_x f(x_t, y_t)\| \leq \|\nabla f(x_t, y_t)\| \leq G$. Therefore, we have

$$\mathbf{E}\mathcal{E}_1 = -\eta_x n \mathbf{E}\langle \nabla_x f(x_t, y_t), \frac{\hat{g}_t}{n} - \frac{g_t}{n}\rangle$$
$$\leq \eta_x \frac{\mu n}{2} L_x (d_x + 3)^{3/2} G.$$

Similarly, we have

$$\mathbf{E}\mathcal{E}_2 = \eta_x^2 L_x n^2 \mathbf{E}\|\frac{\hat{g}_t}{n} - \frac{g_t}{n}\|^2$$
$$\leq \eta_x^2 L_x n^2 \left[32 d_x \|\nabla_x f(x_t, y_t)\|^2 + 108\mu^2 L^2 d^4\right].$$

We further simply the inequality by letting $\eta_x \leq \frac{1}{384 L_x nd}$. Then we have

$$f(x_{t+1}, y_{t+1}) - f(x_t, y_t) + [\nabla f(x_t, y_t)]^\top \begin{bmatrix} \frac{n}{4}\eta_x I_{d_x} & 0 \\ 0 & \frac{n}{3}\eta_y I_{d_y} \end{bmatrix} \nabla f(x_t, y_t)$$
$$\leq \frac{\sigma^2}{2}n^2 \left[\eta_y^3 L_{y,\max}^2 + \eta_x^3 L_{x,\max}^2\right] + o(\mu),$$

where $o(\mu)$ represents a small error term when $\mu$ tends to 0. Lastly, we sum over $t < \tau$ and obtain

$$f(x_\tau, y_\tau) - f^* + \sum_{t<\tau}[\nabla f(x_t, y_t)]^\top \begin{bmatrix} \frac{n}{4}\eta_x I_{d_x} & 0 \\ 0 & \frac{n}{3}\eta_y I_{d_y} \end{bmatrix} \nabla f(x_t, y_t)$$
$$\leq f_0 - f^* + \left[\frac{\sigma^2}{2}n^2 \left[\eta_y^3 L_{y,\max}^2 + \eta_x^3 L_{x,\max}^2\right] + o(\mu)\right] T,$$

which completes the proof. Here, $o(\mu) \leq 3\eta_x \mu n L_x dG$ by letting $\mu \leq \frac{G}{L_x}\frac{6}{d_x^{3/2}}$. $\qquad \square$

## D   PROOF OF THEOREM 1

Here, we re-state our main theorem with full details.

**Theorem 2.** *Suppose that Assumption 1 and Assumption 2 hold for the objective function $f(x, y) := \frac{1}{n}\sum_{i=1}^{n} f(x, y; i)$ and satisfy the hybrid generalized smoothness properties. Let $\delta \in (0, 1)$ and $\{(x_t, y_t)\}_{t=1}^{T}$ be the SGD with Random Shuffling dynamic generated by Algorithm 1 for solving the optimization problem Eq. (1). Given $F$ as*

$$F = \frac{8}{\delta}[f_0 - f^* + \sigma'],$$

*where $f_0 := f(x_0, y_0)$ is the initial function value and $\sigma'$ is a constant-level value given by Eq. (4) and $H$ as*

$$H = 2\sqrt{\frac{[200G^2\frac{d_x}{n} + G^2 + \frac{\sigma^2}{n}]T}{\delta}},$$

*define the stopping time as $\tau = \tau_1 \wedge \tau_2$, where $\tau_1 := \min_t\{t \mid f(x_{t+1}, y_{t+1}) - f^* > F\} \wedge T$ and $\tau_2 := \min_t\{t \mid \|\epsilon_t\| > H\} \wedge T$, where $\epsilon_t$ is defined in Lemma 6. If learning rates $\eta_x$, $\eta_y$, and the perturbation stepsize $\mu$ are chosen such that*

$$\eta_x \leq \min\left\{\frac{1}{2L_{x,\max}n}, \frac{1}{384L_x nd}, \sqrt{\frac{2}{T}}\frac{1}{\sigma n L_{x,\max}}\right\},$$

$$\eta_y \leq \min\left\{\frac{1}{2L_{y,\max}n}, \sqrt{\frac{2}{T}}\frac{1}{\sigma n L_{y,\max}}\right\}, \tag{3}$$

$$\mu \leq \min\left\{\frac{G}{L_x}\frac{6}{d^{3/2}}, \frac{1}{3L_x TndG}\right\}.$$

*where all constant $G, L_{x,\max}, L_{y,\max}, L_x, L_y$ are defined relying on $F$ with presented in Lemma 4, and the maximum number of epoch $T$ is chosen as*

$$T \geq \epsilon^{-2}\left[\frac{2}{\delta} + \frac{G^2}{8}\right] + \epsilon^{-4}\left[\frac{f_0 - f^* + 3}{n}\right],$$

*then with the probability at least $1 - \delta$,*

$$\frac{1}{T}\sum_{t<T}\mathbf{E}\|\nabla f(x_t, y_t)\|^2 \leq \epsilon^2.$$

*Proof.* Let $A := \left\{\frac{1}{T}\sum_{t<T}\|\nabla f(x_t, y_t)\|^2 \leq \epsilon^2\right\}$ and $B := \{\tau \geq T\}$ be two events. We consider the following lower bound of the probability of event $A$ by conditioning it on the event $B$:

$$\mathbb{P}(A) \geq \mathbb{P}(A \cap B) = \mathbb{P}(A|B)\mathbb{P}(B)$$
$$\geq [1 - \mathbb{P}(A^c|B)][1 - \mathbb{P}(B^c)].$$

Our goal is to show that the probability of $\left\{\frac{1}{T}\sum_{t<T}\|\nabla f(x_t, y_t)\|^2 > \epsilon^2\Big|\tau \geq T\right\}$ (the event $A^c|B$) and $\{\tau < T\}$ (the event $B^c$) are both small. We bound each term separately.

- First, we bound the probability of $\left\{\frac{1}{T}\sum_{t<T}\|\nabla f(x_t, y_t)\|^2 > \epsilon^2\Big|\tau \geq T\right\}$. By Lemma 6, we let

$$\sigma' = \left[\frac{\sigma^2}{2}n^2\left[\eta_y^3 L_{y,\max}^2 + \eta_x^3 L_{x,\max}^2\right] + o(\mu)\right]T. \tag{4}$$

If the event is conditioned on $\tau \geq T$, we always have $\|\nabla f(x_t)\| \leq G$ for $t = 1, 2, \ldots, T-1$, where $G$ is determined by Lemma 3. Then we obtain

$$\mathbb{P}\left(\sum_{t<T}\|\nabla f(x_t, y_t)\|^2 > c\Big|\tau \geq T\right) \overset{(i)}{\leq} \mathbb{P}\left(e^{\sum_{t<T}\|\nabla f(x_t, y_t)\|^2} > e^c\Big|\tau \geq T\right)$$

$$\overset{(ii)}{\leq} \mathbf{E}\left[e^{\sum_{t<T}\|\nabla f(x_t, y_t)\|^2}\Big|\tau \geq T\right]/e^c$$

$$\overset{(iii)}{\leq} \exp\left(\sum_{t<T}\mathbf{E}\|\nabla f(x_t)\|^2 + \frac{G^2}{8}\right)/e^c$$

$$\overset{(iv)}{\leq} \exp\left(\frac{1}{\eta_{\min}n}[f_0 - f^* + \sigma'] + \frac{G^2}{8} - c\right).$$

where (i) takes exponential on both sides, (ii) applies the Markov inequality, (iii) applies the Hoeffding's lemma, (iv) applies Lemma 6 with setting $\eta_{\min} = \min\{\frac{\eta_x}{4}, \frac{\eta_y}{3}\}$ and $f_0 := f(x_0, y_0)$.

Before we evaluate the necessary $T$, we need to choose hyper-parameters to make $\sigma'$ less than some constant independent of $d$, $n$, or other crucial constants. To do so, we set

$$\eta_x \leq \sqrt{\frac{2}{T}}\frac{1}{\sigma n L_{x,\max}}, \quad \eta_y \leq \sqrt{\frac{2}{T}}\frac{1}{\sigma n L_{y,\max}}, \quad \mu \leq \frac{1}{3L_x Tnd_x G}.$$

Then we obtain $\sigma' \leq 2\eta_x + \eta_y$. Let $c = T\epsilon^2$ and $e^{\frac{1}{\eta_{\min}n}[f_0 - f^* + 2\eta_x + \eta_y] + \frac{G^2}{8}} e^{-c} \leq \frac{\delta}{2}$. Then it solves

$$\epsilon^2 T \geq \ln(\frac{2}{\delta}) + \frac{G^2}{8} + \frac{1}{\eta_{\min}n}[f_0 - f^* + 2\eta_x + \eta_y]$$

$$T \geq \epsilon^{-2}\left[\frac{2}{\delta} + \frac{G^2}{8}\right] + \epsilon^{-2}\left[\frac{f_0 - f^* + 2\eta_x + \eta_y}{\eta_{\min}n}\right].$$

- Then we bound the probability $\mathbb{P}(B^c) = \mathbb{P}(\tau < T)$. Recap that we consider the stopping time defined as $\tau = \tau_1 \wedge \tau_2$, where $\tau_1 := \min_t\{t \mid f(x_{t+1}, y_{t+1}) - f^* > F\} \wedge T$ and $\tau_2 := \min_t\{t \mid \|\epsilon_t\| > H\} \wedge T$. Here, $\epsilon_t$ is defined as

$$\epsilon_t = \underbrace{\frac{1}{n}\sum_{i=1}^n \hat{\nabla}f(x_{t,i}; \xi_{t,i}) - \frac{1}{n}\sum_{i=1}^n \nabla f(x_{t,i}; \xi_{t,i})}_{\text{est. err.}} + \underbrace{\frac{1}{n}\sum_{i=1}^n \nabla f(x_{t,i}; \xi_{t,i}) - \nabla f(x_t)}_{\text{stoc. err.}}. \quad (5)$$

We note that for the last $d_y$ entries, the estimation error term is $0$ since we do not apply gradient estimation for this part. Both $F$ and $H$ in the definition of stopping times will be determined later. Then we notice that

$$\mathbb{P}(B^c) = \mathbb{P}(\tau < T) = \mathbb{P}(\{\tau_1 < T\} \cup \{\tau_2 < T\})$$
$$= \mathbb{P}(\tau_2 < T) + \mathbb{P}(\tau_1 < T, \tau_2 \geq T).$$

We bound each term separately as follows:

○ Choose $H$ such that $\mathbb{P}(\tau_2 < T) \leq \frac{\delta}{4}$: We have

$$\mathbb{P}(\tau_2 < T) = \mathbb{P}\left(\bigcup_{t<T}\{\|\epsilon_t\| > H\}\right)$$

$$\leq \sum_{t<T}\mathbb{P}\left(\|\epsilon_t\| > H\right)$$

$$\overset{(i)}{\leq} \sum_{t<T}\frac{\frac{3}{n^2}\mathbf{E}\|g_t - \hat{g}_t\|^2 + 3\mathbf{E}\|\frac{g_t}{n} - \nabla_x f(x_t, y_t)\|^2 + 3\mathbf{E}\|\frac{h_t}{n} - \nabla_y f(x_t, y_t)\|^2}{H^2}$$

$$\overset{(ii)}{\leq} \left[\frac{3}{n}\left[64d\|\nabla_x f(x_t, y_t)\|^2 + 216\mu^2 L_{x,\max}^2 d_x^4\right]/H^2 \right.$$

$$\left. + \left(3L_{x,\max}^2\eta_x^2 + 3L_{y,\max}^2\eta_y^2\right)\left[n^2G^2 + n\sigma^2\right]/H^2\right]T$$

$$\overset{(iii)}{\leq} \frac{\left[200G^2\frac{d_x}{n} + 2G^2 + \frac{\sigma^2}{n}\right]T}{H^2}$$

where (i) applies the Markov inequality, (ii) applies Lemma 5 and Lemma 5 from Mishchenko et al. (2020), and (iii) we choose a sufficiently small $\mu \leq \frac{8G}{L_{x,\max}d_x^{3/2}}$ and learning rates $\eta_x \leq \frac{1}{\sqrt{3}L_{x,\max}n}$ and $\eta_y \leq \frac{1}{\sqrt{3}L_{y,\max}n}$ to simplify the upper bound. Then we choose $\frac{\left[200G^2\frac{d_x}{n} + 2G^2 + \frac{\sigma^2}{n}\right]T}{H^2} = \frac{\delta}{4}$. It solves

$$H = 2\sqrt{\frac{[200G^2\frac{d_x}{n} + G^2 + \frac{\sigma^2}{n}]T}{\delta}}. \quad (6)$$

○ Choose $F$ such that $\mathbb{P}(\tau_1 < T, \tau_2 \geq T) \leq \frac{\delta}{4}$. Because $\{\tau_1 < T, \tau_2 \geq T\} \subset \{f(x_\tau, y_\tau) - f^* > \frac{F}{2}\}$,

$$\mathbb{P}(\tau_1 < T, \tau_2 \geq T) \leq \mathbb{P}(f(x_\tau, y_\tau) - f^* > \frac{F}{2})$$

$$\overset{(i)}{\leq} 2\mathbf{E}[f(x_\tau, y_\tau) - f^*]/F$$

$$\leq 2[f_0 - f^* + \sigma']/F.$$

where (i) applies the Markov inequality. Let $\frac{\delta}{4} = 2[f(x_0) - f^* + \sigma']/F$. It solves

$$F = \frac{8}{\delta}[f(x_0) - f^* + \sigma'].$$

(7)

Combining both upper bounds with choosing $H$ and $F$ defined by Eq. (6) and Eq. (7), respectively, we have

$$\mathbb{P}(B^c) = \mathbb{P}(\tau < T) \leq \frac{\delta}{2}.$$

Then we obtain the lower bound of $\mathbb{P}(A \cap B)$ as follows:

$$\mathbb{P}(A \cap B) = \mathbb{P}(A|B)\mathbb{P}(B) \geq [1 - \mathbb{P}(A^c|B)][1 - \mathbb{P}(B^c)]$$

$$\geq [1 - \frac{\delta}{2}][1 - \frac{\delta}{2}] = 1 - \delta + \frac{\delta^2}{4}$$

$$\geq 1 - \delta.$$

Lastly, we discuss the hyper-parameter choices and the epoch complexity. To make Lemma 6 hold, we have set $\eta_x \leq \min\{\frac{1}{2L_{x,\max}n}, \frac{1}{384L_x n d_x}\}$ and $\eta_y \leq \frac{1}{2L_{y,\max}n}$ and the perturbation stepsize $\mu \leq \frac{G}{L_x} \frac{6}{d_x^{3/2}}$. When bounding the probability of $\left\{\frac{1}{T}\sum_{t<T} \|\nabla f(x_t, y_t)\|^2 > \epsilon^2 \middle| \tau \geq T\right\}$ and the probability of $\mathbb{P}(\tau < T)$, we additionally require

$$\eta_x \leq \min\{\sqrt{\frac{2}{T}}\frac{1}{\sigma n L_{x,\max}}, \frac{1}{\sqrt{3}L_{x,\max}n}\},$$

$$\eta_y \leq \min\{\sqrt{\frac{2}{T}}\frac{1}{\sigma n L_{y,\max}}, \frac{1}{\sqrt{3}L_{y,\max}n}\},$$

$$\mu \leq \min\{\frac{1}{3L_x T n d_x G}, \frac{8G}{L_{x,\max}d_x^{3/2}}\}.$$

Therefore, in summary, we have

$$\eta_x \leq \min\left\{\frac{1}{2L_{x,\max}n}, \frac{1}{384L_x n d_x}, \sqrt{\frac{2}{T}}\frac{1}{\sigma n L_{x,\max}}\right\},$$

$$\eta_y \leq \min\left\{\frac{1}{2L_{y,\max}n}, \sqrt{\frac{2}{T}}\frac{1}{\sigma n L_{y,\max}}\right\},$$

$$\mu \leq \min\left\{\frac{G}{L_x}\frac{6}{d_x^{3/2}}, \frac{1}{3L_x T n d_x G}\right\}.$$

Under these hyper-parameter choices, we also need to require

$$T \geq \epsilon^{-2}\left[\frac{2}{\delta} + \frac{G^2}{8}\right] + \epsilon^{-2}\left[\frac{f_0 - f^* + 2\eta_x + \eta_y}{\eta_{\min}n}\right],$$

where $\eta_{\min} = \min\{\frac{\eta_x}{4}, \frac{\eta_y}{3}\}$, to ensure that the probability of $\left\{\frac{1}{T}\sum_{t<T}\|\nabla f(x_t, y_t)\|^2 > \epsilon^2 \middle| \tau \geq T\right\}$ is small (less than $\frac{\delta}{2}$). We observe that by simply setting $\eta_{\min} \leq \epsilon^2$ (we can always make it by choosing $T \geq \Theta(\frac{\epsilon^{-4}}{n^2})$), the above condition on $T$ degenerates to $T \geq \Theta(\frac{\epsilon^{-4}}{n})$. Therefore, it concludes that if $T = \Theta(\epsilon^{-4}/n)$, with the probability at least $1 - \delta$,

$$\frac{1}{T}\sum_{t<T}\|\nabla f(x_t)\|^2 \leq \epsilon^2.$$

Then the proof is completed.

Here, we discuss how we determine the optimal value $\eta_{\min} = \Theta(\epsilon^2)$. In general, we can set $\eta_{\min} = \Theta(\epsilon^\alpha)$, which leads to the condition on $T$: $T \geq \Theta(\epsilon^{-2-\alpha})$. A smaller $\alpha$ is always better. However, we need to ensure the learning rate condition is satisfied; that is, $\eta_{\min} \leq \Theta(\sqrt{\frac{1}{T}})$. It solves $T \leq \Theta(\epsilon^{-2\alpha})$. We let $\epsilon^{-2\alpha} \geq \epsilon^{-2-\alpha}$, which solves $\alpha \geq 2$. Therefore, when $\eta_{\min} = \Theta(\epsilon^2)$, the complexity is optimal and attainable. $\qquad\square$

| | | SST2 | | Copa | | WinoGrande | |
|---|---|---|---|---|---|---|---|
| | | steps | $\mu$ | steps | $\mu$ | steps | $\mu$ |
| **Llama-2-7b** | ZO-FT | $1.1 \times 10^4$ | $10^{-5}$ | $1.6 \times 10^4$ | $10^{-4}$ | $1.8 \times 10^4$ | $10^{-5}$ |
| | FO-Prompt | $6 \times 10^3$ | / | $9 \times 10^3$ | / | $9 \times 10^3$ | / |
| | Hybrid-Prompt | $1.5 \times 10^3$ | $10^{-5}$ | $5 \times 10^3$ | $10^{-5}$ | $9 \times 10^3$ | $10^{-5}$ |
| | FO-Prefix | $2 \times 10^4$ | / | $1.5 \times 10^4$ | / | $3 \times 10^3$ | / |
| | Hybrid-Prefix | $9.5 \times 10^3$ | $10^{-5}$ | $7.5 \times 10^3$ | $10^{-5}$ | $9 \times 10^3$ | $10^{-5}$ |
| | FO-Lora | $2 \times 10^4$ | / | $2.5 \times 10^3$ | / | $2.5 \times 10^3$ | / |
| | Hybrid-Lora | $1.6 \times 10^4$ | $10^{-5}$ | $1.15 \times 10^4$ | $10^{-5}$ | $4.5 \times 10^3$ | $10^{-5}$ |
| **Vicuna-7b-v1.5** | ZO-FT | $1.0 \times 10^4$ | $10^{-5}$ | $7 \times 10^3$ | $10^{-5}$ | $1.75 \times 10^4$ | $10^{-5}$ |
| | FO-Prompt | $2 \times 10^4$ | / | $1.3 \times 10^4$ | / | $2 \times 10^4$ | / |
| | Hybrid-Prompt | $2 \times 10^3$ | $10^{-5}$ | $1.5 \times 10^3$ | $10^{-5}$ | $2 \times 10^4$ | $10^{-6}$ |
| | FO-Prefix | $2 \times 10^3$ | / | $2 \times 10^4$ | / | $2 \times 10^4$ | / |
| | Hybrid-Prefix | $2 \times 10^4$ | $10^{-5}$ | $1.7 \times 10^4$ | $10^{-5}$ | $4 \times 10^3$ | $10^{-5}$ |
| | FO-Lora | $2 \times 10^3$ | / | $9 \times 10^3$ | / | $3.5 \times 10^3$ | / |
| | Hybrid-Lora | $2 \times 10^4$ | $10^{-5}$ | $2.5 \times 10^3$ | $10^{-5}$ | $3 \times 10^3$ | $10^{-5}$ |
| **OPT-1.3b** | ZO-FT | $2 \times 10^4$ | $10^{-5}$ | $8.5 \times 10^3$ | $10^{-4}$ | $8 \times 10^3$ | $10^{-5}$ |
| | FO-Prompt | $2 \times 10^4$ | / | $1.6 \times 10^4$ | / | $9.5 \times 10^3$ | / |
| | Hybrid-Prompt | $2 \times 10^4$ | $10^{-5}$ | $2 \times 10^4$ | $10^{-5}$ | $1.4 \times 10^4$ | $10^{-5}$ |
| | FO-Lora | $3 \times 10^3$ | / | $1.9 \times 10^4$ | / | $1.45 \times 10^4$ | / |
| | Hybrid-Lora | $4 \times 10^3$ | $10^{-5}$ | $1.9 \times 10^4$ | $10^{-5}$ | $3 \times 10^3$ | $5 \times 10^{-4}$ |
| | FO-Prefix | $2 \times 10^4$ | / | $2 \times 10^4$ | / | $9.5 \times 10^3$ | / |
| | Hybrid-Prefix | $8.5 \times 10^3$ | $10^{-5}$ | $1.15 \times 10^4$ | $10^{-5}$ | $2 \times 10^4$ | $10^{-5}$ |

Table 5: A detailed breakdown of the optimal hyperparameters including training steps and $\mu$ specified in Eq. (2) and training specifics for each fine-tuning method applied to different model architectures across SST2, Copa, and WinoGrande tasks. Highlighted cells indicate efficient training processes, showcasing the reduced steps required by hybrid approaches to achieve optimal performance.

# E   EXPERIMENTAL DETAILS

In this paper, we evaluate our proposed hybrid-tuning method across a diverse spectrum of scenarios including three distinct tasks, three transformer-based language models, and three PEFT methods. This extensive exploration not only demonstrates the broad applicability of our approach but also provides robust evidence for its effectiveness and versatility in enhancing model performance across various domains and architectures. In this section, we will briefly review these components and delve into more details of our experiment settings.

## E.1   OVERVIEW OF TASKS

In this section, we briefly discuss the task we consider in our paper. All of tasks are ready to use in the ZO-Bench code base Zhang et al. (2024) and we follow the default setting and the same train/test/validation split of their original implementations.

**Text Binary Classification**   In this paper, we consider the Stanford Sentiment Treebank v2 (SST2) dataset Socher et al. (2013) and the Word-In-Context (WIC) dataset Pilehvar & Camacho-Collados (2018), which presents the simplest binary text classification problem. The SST2 dataset is sufficiently simple and convenience to use to verify our motivating examples (as demonstrated in Figure 1a and Figure 1b). The WIC dataset provides a more challenging task that requires understanding word meanings in different contexts. Both datasets serve as excellent benchmarks for evaluating the performance of our proposed methods in binary text classification tasks.

**Question Answering**  The Choice Of Plausible Alternatives (COPA) dataset Roemmele et al. (2011) is a common benchmark for evaluating the commonsense causal reasoning ability of a language model. It contains one thousand English-language questions answer pairs. We choose this task to evaluate our approaches in improving the question-answering capabilities of models, particularly in scenarios requiring causal inference and commonsense reasoning.

**Common Sense Reasoning Task**  We consider the WinoGrande dataset Sakaguchi et al. (2021) and the Winograd Schema Challenge (WSC) dataset Levesque et al. (2012), which present a challenging common sense reasoning task. The WSC dataset is designed to evaluate machine understanding and reasoning capabilities by presenting pronoun disambiguation problems that require human-like inference. The WinoGrande dataset is designed to be a more difficult and larger-scale version of the original Winograd Schema Challenge, requiring models to demonstrate human-like reasoning capabilities. By including WSC and WinoGrande in our experiments, we aim to assess how well our approaches can enhance a model's ability to reason about complex scenarios and make appropriate inferences based on contextual information.

### E.2 OVERVIEW OF PEFT MODULES

In this paper, we mainly consider three types of PEFT modules. In our proposed hybrid-tuning approach, we jointly train one of these PEFT modules with the base LLM to improve the convergence and overall performance. The following paragraphs provide a detailed overview of the three main PEFT modules considered in this study: Prompt Tuning, Prefix Tuning, and Low-Rank Adaptation (LoRA). In our experiments, we follow the default configuration of Zo-Bench code base Zhang et al. (2024) without making additional modifications. It is worth noting that our hybrid-tuning methods are also applicable to other recently developed PEFT techniques including (1) other LoRA variants such as X-LoRA Buehler & Buehler (2024), Llama-Adapter Zhang et al. (2023b), AdaLoRA Zhang et al. (2023a), LoHa Hyeon-Woo et al. (2021), and LoKr Yeh et al. (2023); (2) other soft prompts techniques such as P-tuning Liu et al. (2021; 2023); and (3) Infused Adapter by Inhibiting and Amplifying Inner Activation (IA3) methods Liu et al. (2022).

**Prompt Tuning**  Prompt tuning Lester et al. (2021) is a lightweight fine-tuning method that prepends trainable continuous prompt tokens to the input. These prompt tokens are optimized during training while keeping the pre-trained language model parameters frozen. This approach allows for task-specific adaptation with a small number of parameters. Prompt tuning is particularly effective for large language models and can be seen as a form of soft prompting that learns optimal input representations for specific tasks.

**Prefix Tuning**  Prefix tuning Li & Liang (2021) extends the concept of prompt tuning by adding trainable prefix tokens not only to the input but to each layer of the transformer model. This method prepends a trainable continuous prefix to the keys and values of the self-attention layers in each transformer block. By doing so, prefix tuning allows for more flexible and expressive task-specific adaptations compared to prompt tuning, while still maintaining a relatively small number of trainable parameters.

**LoRA**  Low-Rank Adaptation (LoRA) Hu et al. (2021) is a parameter-efficient fine-tuning method that adds low-rank decomposition matrices to the weights of the pre-trained model. Instead of directly updating the model's weight matrices, LoRA introduces pairs of rank decomposition matrices for each weight matrix being tuned. These low-rank matrices are initialized randomly and trained to adapt the model to specific tasks. LoRA significantly reduces the number of trainable parameters while maintaining competitive performance compared to full fine-tuning. It offers several advantages, including faster training, lower memory requirements, and the ability to switch between multiple fine-tuned tasks efficiently by changing only the LoRA parameters.

### E.3 CONVERGENCE OF HYBRID FINE-TUNING

In this subsection, we present the training curves (including the training loss, validation accuracy, and the test accuracy) for OPT-1.3B Zhang et al. (2022) model on SST-2 Socher et al. (2013) dataset in Figure 4. We observe that a significant efficiency gain in terms of training steps. The hybrid

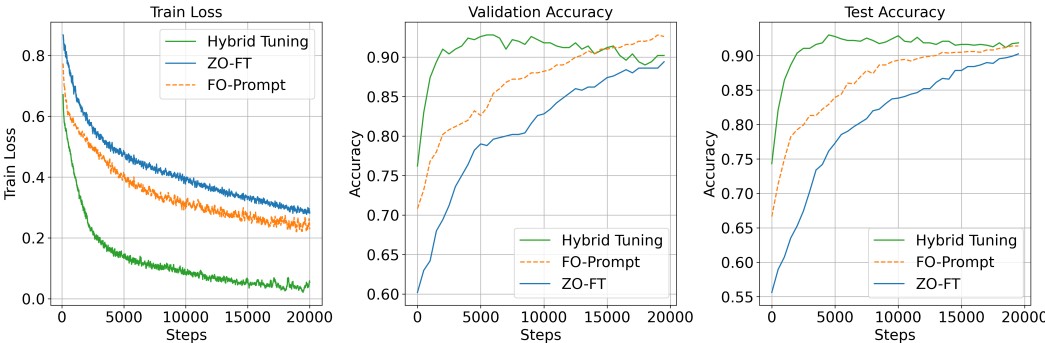

(a) Training curves for OPT-1.3B model with the prompt tuning on the SST2 dataset with using the optimal hyper-parameter indicated in Table 5. The hybrid-tuning achieves the significant better performance. Notably, this phenomenon is also observed in other tasks and for other model architectures.

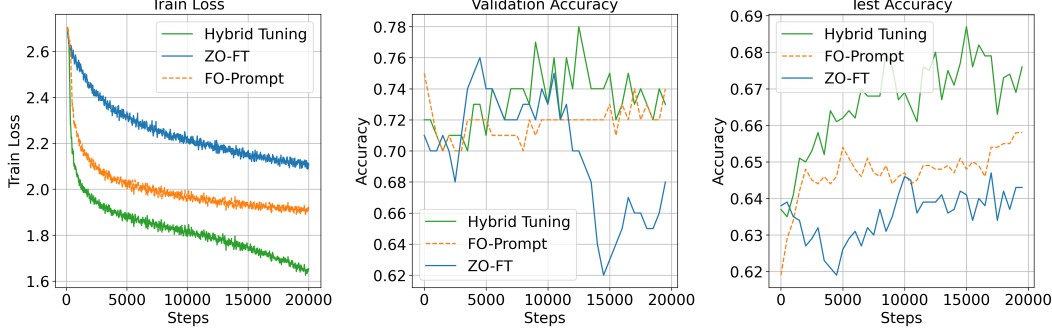

(b) Training curves for Vicuna-7b-v1.5 model with the prompt tuning on the WinoGrande dataset.

Figure 6: Comparison of training curves for different models and datasets. These results demonstrate that the similar outperformance of hybrid-tuning is observed across various model architectures and NLP tasks.

method consistently achieves optimal performance regarding the training loss. This trend is observed across different tasks, PEFT methods, and model architectures, suggesting that the efficiency of hybrid tuning scales well (e.g. for Vicuna-7b-v1.5 model on the WinoGrande dataset in Figure 6b). A detailed breakdown of is provided in Table 5.

### E.4 ESTIMATING SMOOTHNESS

In Figure 1a and Figure 1b, the smoothness of the loss landscape of the OPT-125M (and the LoRA module) is estimated by approximating the norm of Hessian matrix at the stochastic data point using the zeroth-order gradient estimation to the Hessian-vector products (HVPs):

$$\text{Hessian}(x)^\top v \approx \sum_{\xi \in \text{Batch}} \frac{\nabla f(x + hv; \xi) - \nabla f(x; \xi)}{h},$$

where $\nabla f(x; \xi)$ is the stochastic gradient at $x$ for the data point $\xi$ in the given data batch, $h$ is a small perturbation size, and $v$ is a random unit vector. We estimate the Frobenius norm $\|\text{Hessian}(x)\|_F \approx \sqrt{\mathbf{E}v^\top H^2 v}$ of the Hessian by sampling multiple random vectors and computing these HVPs.

For Figure 1a, we initialize the parameter of pre-trained binary classification OPT-125M model and train it over the SST2 dataset for 5000 steps with setting the learning rate $\eta = 5 \times 10^{-5}$ and the batch size 8. We sample 100 independent vectors from the unit sphere to estimate the HVP with the perturbation $h = 10^{-5}$ and obtain the Hessian norm as the approximation of the local smoothness constant $L$.

For Figure 1b, we initialize the parameter of pre-trained binary classification OPT-125M model as the base model and randomly initialize the LoRA module with the rank 16 and the LoRA Alpha 32

(the detailed configuration can be found in the source code) and jointly train both components over the SST2 dataset for 5000 steps with setting the learning rate $\eta = 5 \times 10^{-5}$ and the batch size 8.We collect all parameters along the SGD trajectories. We perturb the parameter of the base LLM and the LoRA module, respectively, with 100 independent vectors from the unit sphere and the perturbation $h = 10^{-5}$ to estimate the smoothness.

### E.5 Omitted Experimental Settings

Following the methodology of Malladi et al. (2023); Zhang et al. (2024), we assessed our approach on 6 representative NLP tasks including the sentiment classification task on the SST2 dataset Socher et al. (2013), the sentence differing task on the WSC dataset Levesque et al. (2012), contextualized word and sense representation and word sense disambiguation task on the WiC dataset Pilehvar & Camacho-Collados (2018), the question answering task on the COPA dataset Roemmele et al. (2011), and the common sense reasoning task on the WinoGrande dataset Sakaguchi et al. (2021). The models we use in our experiments include OPT-1.3b Zhang et al. (2022), Vicuna-7b Chiang et al. (2023), and LLaMA-7b Zhang et al. (2023b). We compare the performance of our approach against standard PEFT methods including first-order prompt tuning Lester et al. (2021), LoRA tuning Hu et al. (2021), and prefix tuning Li & Liang (2021). For each dataset, we randomly sample 1,000 examples for training, 1,000 examples for evaluation, and 100 examples for development. Performance is evaluated using accuracy or F1 score, as appropriate for each task. All experiments utilize SGD as the optimizer. In the case of prompt tuning and prefix tuning, the prompts are initialized according to the predefined settings in Table E.2 of Malladi et al. (2023), while for LoRA tuning, we initialize with zeros. We perform hyperparameter tuning for all methods and report the best configurations. For all methods, we set the maximum number of training steps to 20,000, with early stopping applied when applicable. The detailed hyperparameter setting, overviews of the tasks and PEFT methods, hyper-parameter setting, and the full results are reported in the supplementary materials.

For the zeroth-order approximation, we follow the same approach outlined by Malladi et al. (2023). In the case of prompt tuning and prefix tuning, the prompts are initialized according to the predefined settings in Table E.2 of Malladi et al. (2023), while for LoRA tuning, we initialize with zeros. We perform hyperparameter tuning for all methods and report the best configurations. For all methods, we set the maximum number of training steps to 20,000, with early stopping applied when applicable.

### E.6 Hyper-Parameter Searching

In our experiments, we conducted systematic grid searches across all combinations of tasks, models, and PEFT methods. For FO PEFT training configurations, we primarily grid-searched the learning rate (among 0.001, 0.0001, 0.00001, and 0.000001), while maintaining fixed hyperparameters for LoRA (rank=8, alpha=16) and Prompt Tuning (10 virtual tokens). In hybrid training configurations, we adjust the search space to include both the base learning rate (0.001 and 0.0001) and the zero-order (ZO) learning rate ($10^{-6}$ and $10^{-7}$), creating a two-dimensional grid search with the same number of hyper-parameter as the FO method. Notably, our hyperparameter configurations are inspired by theoretical analysis of hybrid fine-tuning (*i.e.* the base LLM requires a smaller learning rate), which allows us to strategically constrain the grid-search space. This theoretical guidance not only reduces computational overhead but also demonstrates how theoretical insights can effectively streamline the practical implementation of fine-tuning procedures.

All experiments maintained consistent training parameters including 5 epochs, batch size of 16, and 20,000 maximum steps, with evaluation performed every 500 steps. The search strategy was implemented using grid search methodology, with accuracy on the validation set as the optimization metric.

## F Extended Discussions on Wall-Clock Time

While standard zero-order methods often accelerate training by bypassing backpropagation (Malladi et al., 2023; Zhang et al., 2024), our Hybrid Tuning maintains comparable efficiency by restricting the expensive first-order updates solely to the PEFT module. This design ensures minimal com-

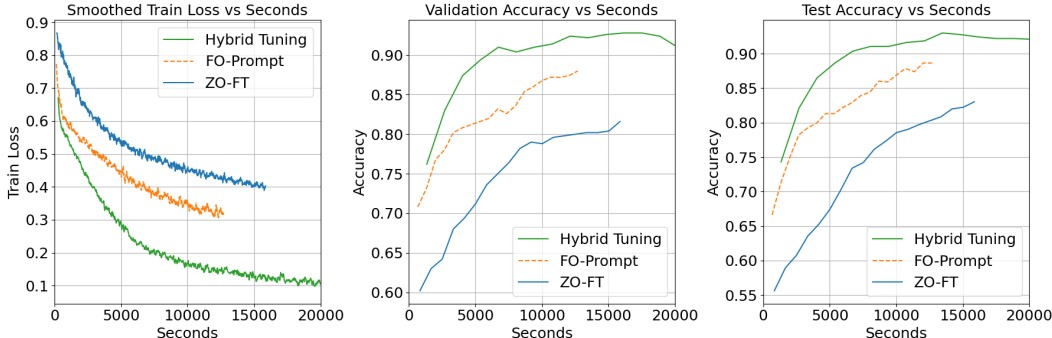

Figure 7: Training performance of OPT-1.3B on SST-2 using prompt tuning versus wall-clock time. The time cost of each method is: Hybrid Tuning (1.28 sec/step), FO-Prompt (0.80 sec/step), ZO-FT (0.64 sec/step), and one forward pass (0.15 sec/step).

putational overhead despite requiring both forward-pass estimation and backpropagation. Empirical tests on Llama-2-7b confirm that our method (2.36 sec/step) remains competitive with full-parameter FO-SGD (2.27 sec/step) (one forward pass costs 0.34 sec/step). To visualize this efficiency, Figure 7 plots the training curve of OPT-1.3B (with prompt tuning) on the SST-2 dataset against wall-clock time, as the comparison against Figure 4.

## G   THEORETICAL CONTRIBUTIONS

On the theoretical side, our work addresses key gaps in the current optimization literature.

- First, we introduce and analyze SGD under a novel *hybrid smoothness condition* (Definition 1), which generalizes both classical $L$-smoothness assumptions and $\ell$-generalized smoothness assumption; this condition better reflects the heterogeneous structure of modern hybrid models. To our best knowledge, this is the first formal treatment of SGD under such a condition.

- Second, we extend the analysis to the *random reshuffling* setting, marking the first convergence result that integrates generalized smoothness with reshuffling-based SGD algorithms.

- Finally, we improve the known sample complexity bounds for SGD under generalized smoothness by applying sharper concentration techniques. This leads to a provable improvement in the dependence on the confidence parameter $\delta$, reducing it from $O(\epsilon^{-4}/\delta)$ to $O(\epsilon^{-2}/\delta + \epsilon^{-4})$.

## H   BROADER IMPACTS

The proposed hybrid fine-tuning framework has the potential to broadly impact the development and deployment of LLMs by addressing both computational efficiency and adaptability to new tasks. By combining ZO optimization for the base LLM with FO optimization for PEFT module, the approach enables scalable and memory-efficient training without sacrificing performance. This hybrid strategy introduces a novel theoretical framework—the hybrid smoothness condition—that rigorously accounts for heterogeneous parameter landscapes, offering insights relevant not only to NLP but also to general large-scale machine learning systems. The framework could facilitate broader accessibility of LLM fine-tuning in resource-constrained environments and inspire future work on hybrid optimization in other domains.

## I   LIMITATIONS

While our hybrid fine-tuning approach demonstrates strong empirical performance and theoretical convergence guarantees, it has several limitations. First, the effectiveness of the method relies on

tuning separate learning rates for the base LLM and PEFT modules, which may require additional hyperparameter search. Second, the ZO optimization used for updating the base model, though more memory-efficient, can still be computationally expensive due to repeated function evaluations, especially for large-scale models. Finally, the current formulation is limited to joint training of LLMs with PEFT modules and may not generalize directly to other forms of model composition, such as mixture-of-agent or other multi-agent systems. Addressing these challenges remains an avenue for future research.

## J    LLM USAGE

We primarily employed a large language model (LLM) as an auxiliary tool to support the preparation of this manuscript. The LLM was used for tasks such as language polishing, improving clarity, and suggesting alternative phrasings. It did not generate new research ideas, perform data analysis, or contribute substantively to the scientific content of the work. All conceptual development, methodology, experimental design, and interpretation of results were carried out independently by the authors. The authors take full responsibility for the content of this paper, including all text revised with the aid of the LLM. The LLM is not considered an author or contributor.

## K    CONCLUSION

In conclusion, this work introduces a novel hybrid fine-tuning approach for LLMs that combines zeroth-order optimization for the base model with first-order optimization for PEFT modules. Motivated by the hybrid smoothness condition of our hybrid fine-tuning system (Definition 1), we develop a theoretical framework centered on this theoretical challenge introduced by the hybrid fine-tuning method. Our empirical examples (Section 2.2) and convergence analysis (Theorem 1) demonstrate the necessity of applying different learning rates for different modules. Our analysis achieves the best-known sample complexity under much milder conditions in the existing literature. Extensive empirical evaluations across multiple NLP tasks, model architectures, and PEFT techniques validate the theoretical insights and show consistent performance gains over traditional fine-tuning methods as shown in Table 1. By addressing fundamental challenges in joint LLM and PEFT training, our work opens new avenues for efficient LLM fine-tuning and provides a solid foundation for future research on optimizing hybrid systems in machine learning.

