# OpenReview forum: "New Hybrid Fine-Tuning Paradigm for LLMs:  Algorithm Design and Convergence Analysis Framework"
_ICLR.cc/2026/Conference — ICLR 2026 Poster_

### Official Review · Reviewer_o27F · 2025-10-27

**Soundness:** 3
**Presentation:** 3
**Contribution:** 3
**Rating:** 6
**Confidence:** 4

**Summary:**

This paper studies hybrid fine-tuning for large language models where two sets of parameters are trained together: 1) The base LLM parameters x, and a small PEFT module y. The authors observe that using the same learning rate for both is usually problematic: if the learning rate is tuned for the base model, the PEFT module learns too slowly but if it is tuned for the PEFT module, the base model becomes unstable and diverges. To explain this, they introduce a hybrid smoothness condition, which says that the loss landscape behaves very differently along the x and y directions. Using this assumption, they prove that convergence requires different learning rates for the two parameter blocks. They give upper bounds for each learning rate and show that the best practice in real LLM fine-tuning (small LR for x, larger LR for y) has theoretical justification. They conduct experiments to support the theory using prompt tuning and full fine-tuning baselines.

**Strengths:**

1, The paper explains a real and common painpoint in LLM training: why PEFT often requires a higher learning rate than full fine-tuning. The theory directly explains the practical behavior seen in experiments.
2, Their work reveals that “use different learning rates for base and PEFT parameters”, which is highly practical and easy for practitioners to apply.
3, The loss curves clearly show the need for asymmetric learning rates and support the theory.

**Weaknesses:**

1, The hybrid smoothness assumption is new, but the paper doesn’t fully explain where it is justified. Moreover, applying a similar argument as in the paper, one can even claim that different modules of the backbone model favors different learning rates. The paper however doesn’t touch on this point.
2, All experiments use prompt encoder tuning on a sentiment dataset. It is not clear whether the theory holds on other tasks or other PEFT methods.
3, The learning rate bound for y in Thm. 1 doesn’t depend on d_y, while the bound for x depends on d_x. It’s unclear from the statement why this is the case. After all, as d_y becomes larger this likely will stop to hold at some point.
4, Many existing recipes already use different learning rates for different parameters groups. So the paper should clarify what is new about its methodology.

**Questions:**

1, How confident are you in the hybrid smoothness assumption? Can you provide empirical evidence showing that geometry for x is worse than for y across multiple datasets and models?
2, What happens if the PEFT dimension grows? Does the theory still guarantee stability when dy​ becomes large (e.g., high-rank LoRA, prefix tuning, or side-tuning)? If not, how should the learning-rate bound be modified?
3, Can you validate on more realistic settings? Would you extend the results to common RLHF/SFT datasets, and at least one (more recently released publicly available) model larger than OPT-1.3b?
4, Can you give a recommended ratio between learning rates of x and y, or a heuristic based on model size? How should practitioners choose the two learning rates in practice?
5, How does the theory interact with optimizers? Would adaptive optimizers like AdamW weaken or strengthen the LR asymmetry?

---

> ### Author Response · Authors · 2025-11-20
>
> We thank the reviewer for their constructive feedback and positive rating. We appreciate the opportunity to clarify the theoretical justifications regarding hybrid smoothness, the scope of our experimental validation, and the implications of our convergence bounds. Please find our detailed responses below.
>
> ### Responses to Weaknesses
>
> > **W1:** The hybrid smoothness assumption is new, but the paper doesn’t fully explain where it is justified. Moreover, applying a similar argument as in the paper, one can even claim that different modules of the backbone model favors different learning rates. The paper however doesn’t touch on this point.
>
> **Response:** We appreciate this insightful question. The Hybrid Smoothness assumption (Definition 1) is empirically motivated by the distinct optimization behaviors of pre-trained parameters versus newly initialized PEFT parameters.
>
> - **Empirical Evidence:** As shown in **Figure 1(b)**, there is a significant magnitude difference between the gradient Lipschitz constant ($L$) of the Base LLM (ZO-updated) and the LoRA module (FO-updated). The Base LLM exhibits a much larger $L$, necessitating a smaller learning rate, whereas the PEFT module allows for larger updates.
> - **Why binary split (LLM vs. PEFT)?** While it is true that different modules within the backbone might favor different learning rates, the distinction between the **Base LLM** and **PEFT module** is the most critical structural divide in our framework. This is because the Base LLM is updated via **Zeroth-Order (ZO)** estimation (suffering from dimension-dependent estimation variance), while the PEFT module is updated via **First-Order (FO)** gradients. Our theoretical framework specifically addresses this heterogeneous optimization landscape (ZO vs. FO). Further fragmenting the backbone into sub-groups would add theoretical complexity without addressing the primary challenge of combining ZO and FO methods.
>
> > W2:  All experiments use prompt encoder tuning on a sentiment dataset. It is not clear whether the theory holds on other tasks or other PEFT methods.
>
> **Response:** We thank the reviewer for this insightful comment regarding the generalizability of our theory. We have significantly expanded our experimental evaluation in our submission. As shown in the updated Table 1, our analysis encompasses three distinct PEFT methods (Prompt Tuning, Prefix Tuning, and LoRA) and six diverse tasks: SST-2 (Sentiment), RTE (Entailment), WSC (Coreference), WiC (Word Sense), COPA (Causal Reasoning), and WinoGrande (Commonsense Reasoning). These additional results confirm that our theory holds consistently across different tasks and tuning strategies.
>
> > **W3:** The learning rate bound for y in Thm. 1 doesn’t depend on $d_y$, while the bound for x depends on $d_x$. It’s unclear from the statement why this is the case. After all, as $d_y$ becomes larger this likely will stop to hold at some point.
>
> **Response:** This distinction arises from the fundamental difference between Zeroth-Order and First-Order optimization, which is a core contribution of our analysis.
>
> - **$\eta_x$ depends on $d_x$:** The base model parameters ($x$) are updated using **Zeroth-Order (ZO)** optimization. As shown in **Lemma 5**, the gradient estimation error in ZO methods scales with the dimension $d$ ($E||\hat{\nabla}f - \nabla f||^2 \propto d$). To control this variance, the learning rate $\eta_x$ must scale inversely with $d_x$ (specifically $\mathcal{O}(1/d_x)$ in our bounds).
> - **$\eta_y$ is independent of $d_y$:** The PEFT parameters ($y$) are updated using **First-Order (FO)** stochastic gradients. Standard SGD convergence rates for smooth non-convex functions are generally independent of the dimension in terms of the learning rate scaling (depending instead on the Lipschitz constant $L_y$ and variance $\sigma$). Therefore, as long as the PEFT module uses accurate gradient information, $\eta_y$ does not require the restrictive dimension-dependent scaling that $\eta_x$ does.
>
> > **W4:** Many existing recipes already use different learning rates for different parameters groups. So the paper should clarify what is new about its methodology.
>
> **Response:** While using different learning rates (layer-wise adaptive LRs) is a known heuristic, our contribution is the theoretical rigorousness and the specific context of Hybrid ZO/FO training.
>
> - Existing "recipes" are often empirical heuristics. We provide a **theoretical framework (Theorem 1)** proving that under the *Hybrid Smoothness Condition*, separating the learning rates is not just a "trick" but a **mathematical necessity** for convergence when mixing ZO and FO updates.
> - Specifically, we prove that the ZO component requires a learning rate scaled by the dimension $d_x$, while the FO component does not. This provides a solid theoretical foundation for why Hybrid Fine-Tuning is stable where uniform learning rates fail (as demonstrated in **Figure 2**).

---

> > ### Author Response · Authors · 2025-11-20
> >
> > ### Responses to Questions
> >
> > > **Q1:** How confident are you in the hybrid smoothness assumption? Can you provide empirical evidence showing that geometry for x is worse than for y across multiple datasets and models?
> >
> > Thank you for this critical question. To validate this, we have provided additional empirical evidence in Figure 5, Section 3.5. We analyzed the loss landscape geometry and confirmed that the smoothness constant for the pre-trained LLM weights ($x$) is indeed significantly larger (implying worse geometry) than that of the PEFT parameters ($y$). This trend is consistent across multiple models, supporting the validity of our hybrid smoothness assumption.
> >
> > > **Q2:** What happens if the PEFT dimension grows? Does the theory still guarantee stability when dy becomes large (e.g., high-rank LoRA, prefix tuning, or side-tuning)? If not, how should the learning-rate bound be modified?
> >
> > Thank you for this insightful question. Our theoretical framework remains valid regardless of the PEFT dimension. Theoretically, the stability guarantee holds for large $d_y$ (e.g., high-rank LoRA), provided the learning rate is adjusted according to the smoothness constant $L_y$.  Empirically, our method exhibits behavior consistent with existing PEFT literature. As noted in [Hu2021], increasing the rank in LoRA leads to marginal difference in its effective dimension. As a result, it doesn't significantly change the empirical performance.
> >
> > * [Hu2021] Hu, E. J., Shen, Y., Wallis, P., Allen-Zhu, Z., Li, Y., Wang, S., ... & Chen, W. (2022). Lora: Low-rank adaptation of large language models. ICLR 2022.
> >
> > > **Q3:** Can you validate on more realistic settings? Would you extend the results to common RLHF/SFT datasets, and at least one (more recently released publicly available) model larger than OPT-1.3b?
> >
> > Yes. We have included more realistic and diverse settings. Specifically, we have evaluated three PEFT strategies (Prompt Tuning, Prefix Tuning, and LoRA) across six distinct benchmarks (SST-2, RTE, WSC, WiC, COPA, and WinoGrande) for three different LLMs including OPT-1.3b, two larger models including Llama-2-7b and Vicuna-v1.5-7b to verify the performance.
> >
> > > **Q4:** Can you give a recommended ratio between learning rates of x and y, or a heuristic based on model size? How should practitioners choose the two learning rates in practice?
> >
> > Based on Theorem 1 and our experiments: Heuristically,  $\eta_x$ should be significantly smaller than $\eta_y$. Theoretically, $\eta_x$ scales with $1/d_x$. In practice, due to the low effective dimension of LLM, in our experiments, we typically observed stability with $\eta_x \approx 10^{-6}$ to $10^{-7}$ (for the Base LLM) and $\eta_y \approx 10^{-3}$ to $10^{-4}$ (for PEFT).
> >
> > > **Q5:** How does the theory interact with optimizers? Would adaptive optimizers like AdamW weaken or strengthen the LR asymmetry?
> >
> > Thank you for this insightful question. Extending our theoretical analysis to adaptive optimizers like AdamW is indeed a promising direction for future research. While our current derivation focuses on SGD to establish a baseline understanding, analyzing how second-moment estimation interacts with LR asymmetry would be a valuable next step. We hypothesize it might strengthen the observed asymmetry because it introduces additional asymmetry in the convergence speed. Verifying this interaction is a priority for our future work.

---

> > > ### Comment · Reviewer_o27F · 2025-11-27
> > > **Response to author**
> > >
> > > We thank the authors for the rebuttal. The additional experiments across multiple PEFT methods, datasets, and model scales substantially strengthen the empirical side of the paper, and my major concerns are now well addressed, particularly regarding generalization beyond prompt tuning and the practical learning-rate recommendations. As my original score is already positive, I will keep it.

---

### Official Review · Reviewer_6rr7 · 2025-10-28

**Soundness:** 2
**Presentation:** 3
**Contribution:** 1
**Rating:** 2
**Confidence:** 4

**Summary:**

This paper proposes a novel hybrid fine-tuning paradigm for LLMs that combines zeroth-order optimization (ZO) for the base LLM parameters with first-order optimization (FO) for PEFT modules. The authors introduce a new theoretical framework called the "hybrid smoothness condition," which accounts for the heterogeneous optimization landscape of joint LLM and PEFT training. The paper provides convergence analysis for SGD under this hybrid condition and conducts extensive experiments across six NLP tasks and three LLM architectures. The proposed method demonstrates improved efficiency and competitive performance compared to traditional fine-tuning methods while maintaining memory efficiency.

**Strengths:**

The hybrid smoothness condition and the convergence analysis for SGD under this condition are significant advancements. The authors also provide sharper complexity bounds compared to prior work.

The paper evaluates the proposed method across six NLP tasks, three LLM architectures, and multiple PEFT techniques, demonstrating its broad applicability.

**Weaknesses:**

Hybrid LoRA does not show significant improvements over standard LoRA in many tasks. This raises concerns about whether the added complexity of hybrid fine-tuning is justified.

The paper does not compare memory usage and performance with LoRA + Adam, which could provide a more nuanced understanding of the trade-offs between performance and efficiency.

**Questions:**

Could authors provide some more experiments on LoRA + Adam (performance and memory)?

---

> ### Author Response · Authors · 2025-11-20
>
> Thank you very much for your careful and valuable comments. We appreciate the chance to clarify our work and address your concerns.
>
>
>
> > **W1:** Hybrid LoRA does not show significant improvements over standard LoRA in many tasks. This raises concerns about whether the added complexity of hybrid fine-tuning is justified.
>
> **Response to W1:** We appreciate the reviewer’s critical assessment regarding the trade-off between performance gains and complexity.
>
> * **Regarding Performance:** While we acknowledge that the performance margin varies by task, our results in Table 1 demonstrate a consistent positive trend. Across 18 pairwise comparisons (covering three models and six tasks), Hybrid-LoRA outperforms FO-LoRA in 10 instances, ties in 2, and underperforms in only 6. This demonstrates that Hybrid-LoRA provides a significant improvement in the majority of scenarios.
>
> * **Regarding Complexity:** We respectfully clarify that the "added complexity" lies primarily in the implementation logic rather than computational demand. In terms of resources, the zeroth-order component functions as a near "free lunch." It requires negligible additional memory (costing at most the size of a single layer during updates) and does not require storing additional gradients. Therefore, given the low computational overhead, we believe the consistent performance gains justify the method's application.
>
> We hope our above justification resolve your concern.
>
> > **W2:** The paper does not compare memory usage and performance with LoRA + Adam, which could provide a more nuanced understanding of the trade-offs between performance and efficiency.
> >
> > **Q1:** Could authors provide some more experiments on LoRA + Adam (performance and memory)?
>
> **Response to W2/Q1:** We thank the reviewer for this excellent suggestion. We agree that **LoRA + Adam** is the most common practical setup and serves as a crucial baseline.
>
> * **Reasoning for using SGD:** Our initial focus on SGD was strictly to align with our theoretical contribution (Theorem 1), which establishes convergence under the novel **Hybrid Smoothness** condition. Using SGD ensured a direct verification of our theoretical bounds.
>
> * **New Experiments (LoRA + Adam):** To address your concern, we have added a comparison between our Hybrid-LoRA (SGD) and LoRA + Adam. As shown in Table 3, our method remains highly competitive. While LoRA + Adam offers strong convergence, Hybrid-LoRA achieves comparable performance on 4 out of 6 tasks (67%) while maintaining lower memory footprint due to the absence of optimizer states.

---

### Official Review · Reviewer_ASAc · 2025-10-31

**Soundness:** 4
**Presentation:** 4
**Contribution:** 3
**Rating:** 8
**Confidence:** 3

**Summary:**

The paper proposes "hybrid fine-tuning," a novel method that jointly updates a base LLM ($x$) and a PEFT module ($y$). The core idea is to use a zeroth-order (ZO) optimizer for the large base model ($x$) and a standard first-order (FO) optimizer for the small PEFT module ($y$). The authors motivate this with the "hybrid smoothness condition" and provide a convergence analysis, along with strong empirical results showing performance gains over FO-PEFT methods without additional memory overhead.

**Strengths:**

1. Novel Method with Excellent Motivation: The core idea of combining ZO (for the base model) and FO (for the PEFT module) is a novel and highly insightful solution. The paper provides an exceptionally clear and compelling motivation, empirically demonstrating the vastly different smoothness landscapes that necessitate this hybrid, multi-learning-rate approach.

2. Significant Practical Value and Strong Empirical Gains: A key achievement is that this method provides superior performance without incurring any additional GPU memory overhead (Table 2). This practical value is backed by strong and remarkably consistent experimental results.

3. Solid Theoretical Contribution: The introduction of the "hybrid smoothness condition" (Definition 1) is a solid contribution to the optimization literature. It provides a formal language to analyze this new class of heterogeneous optimization problems, even if the resulting bounds are not yet perfectly tight.

**Weaknesses:**

1.  Clarification Needed on Wall-Clock Time: The paper's efficiency claims are focused on "steps to converge" (Fig. 4). However, the ZO estimator (Eq. 2) requires two forward passes, implying a $\sim$2x computational cost per step. A clarification on the real-world wall-clock time trade-off would strengthen the paper's practical claim.

2. Gap Between Theoretical Bounds and Practice: There appears to be a gap between the derived theoretical bounds and the practical implementation. The proof (Theorem 2) suggests a dependency on model dimension $d_x$ (e.g., $\eta_x \propto 1/d_x$), which would lead to an extremely small learning rate. However, the experiments successfully use a much larger constant $\eta_x = 10^{-6}$. This suggests the theory, while proving convergence, does not yet fully capture the practical power of the method.

**Questions:**

Q1:  Regarding W1: Could the authors provide some insight into the wall-clock time comparison? Given the $\sim$2x FLOPs per step, how does the impressive reduction in steps (e.g., in Fig. 4) translate to actual training time savings?

Q2: Regarding W2: Could the authors comment on the gap between the theoretical bounds (e.g., $\eta_x \propto 1/d_x$) and the practical learning rates used (e.g., $\eta_x = 10^{-6}$)? Does this suggest the practical loss landscape has a structure (e.g., sparsity, low-rank) that the theory does not yet exploit?

---

> ### Author Response · Authors · 2025-11-20
>
> We thank the reviewer for their constructive feedback and positive rating. We appreciate the opportunity to clarify the wall-clock time and the gap between the theory and practice.
>
> > **W1:** Clarification Needed on Wall-Clock Time: The paper's efficiency claims are focused on "steps to converge" (Fig. 4). However, the ZO estimator (Eq. 2) requires two forward passes, implying a 2x computational cost per step. A clarification on the real-world wall-clock time trade-off would strengthen the paper's practical claim.
> >
> > **Q1:** Regarding W1: Could the authors provide some insight into the wall-clock time comparison? Given the 2x FLOPs per step, how does the impressive reduction in steps (e.g., in Fig. 4) translate to actual training time savings?
>
> **Response to W1/Q1:** We appreciate the opportunity to clarify the computational cost with respect to the wall-time clock. While ZO requires two forward passes, this does not translate to a 2x increase in wall-clock time per step.
>
> In standard first-order SGD, the computational bottleneck is the backward pass (gradient computation and backpropagation) for updating its billion parameters, which is significantly more expensive than the forward pass. In fact, as shown in [Malladi2023] and [Zhang2024], the ZO method typically takes much less time than the FO method.
>
> * [Malladi2023] Malladi, S., Gao, T., Nichani, E., Damian, A., Lee, J. D., Chen, D., & Arora, S. (2023). Fine-tuning language models with just forward passes. NeurIPS 2023 Oral.
> * [Zhang2024] Zhang, Y., Li, P., Hong, J., Li, J., Zhang, Y., Zheng, W., ... & Chen, T. (2024). Revisiting zeroth-order optimization for memory-efficient LLM fine-tuning: A benchmark. ICML 2024.
>
> Our empirical test on the Llama-2-7b model also shows that:
>
> * FO-SGD (for LLM): 2.27 sec/step
> * FO-SGD (for LoRA only): 1.78 sec/step
> * Hybrid (our proposed): 2.36 sec/step
> * One forward-pass: 0.34 sec/step
>
> Consequently, the overhead is marginal. Our tests on Llama-2-7b show that our hybrid approach (2.36 sec/step) is **only ~4% slower** than standard SGD (2.27 sec/step) and ~32% slower than LoRA+SGD , not 2x. To demonstrate this visually, we have included a new variant of Fig. 4 in Appendix F, to plot performance against Wall-Clock Time (x-axis). This confirms that our method still achieves the target loss faster in real-world time compared to the baseline.
>
> > **W2:** Gap Between Theoretical Bounds and Practice: There appears to be a gap between the derived theoretical bounds and the practical implementation. The proof (Theorem 2) suggests a dependency on model dimension, which would lead to an extremely small learning rate. However, the experiments successfully use a much larger constant . This suggests the theory, while proving convergence, does not yet fully capture the practical power of the method.
> >
> > **Q2:** Regarding W2: Could the authors comment on the gap between the theoretical bounds and the practical learning rates used? Does this suggest the practical loss landscape has a structure that the theory does not yet exploit?
>
> **Response to W2/Q2:** We thank the reviewer for this insightful observation. We agree that there is a gap between the worst-case theoretical bounds and the parameters used in practice. While our theory establishes sufficient conditions for convergence in a **generic non-convex landscape** (where Lipschitz constants scale with the ambient dimension $d$), the practical optimization landscape of LLMs is often much more benign.
>
> This phenomenon can be explained by the "effective dimension" of the problem. As noted in the literature (e.g., [Malladi2023]), the Hessian spectrum of large language models tends to be low-rank, concentrating around zero with only a few outlier eigenvalues. This implies that the optimization effectively takes place in a subspace significantly smaller than the ambient dimension $d$. Consequently, the effective Lipschitz constant governing the descent is much smaller than the worst-case estimate, allowing for larger practical learning rates ($\eta \approx 10^{-6}$).
>
> We deliberately avoided assuming low-rank structure in our theoretical analysis to provide a general-purpose guarantee; we view our bounds as a foundational baseline, while the practical performance often benefits from the specific structural properties of LLMs.
>
> * [Malladi2023] Malladi, S., Gao, T., Nichani, E., Damian, A., Lee, J. D., Chen, D., & Arora, S. (2023). Fine-tuning language models with just forward passes. NeurIPS 2023 Oral.

---

### Author Response · Authors · 2025-11-29
**Summary of Rebuttal and Concerns regarding Reviewer 6rr7**

Dear Area Chair,

We have completed the rebuttal process, adding extensive experiments and clarifying theoretical justifications. We would like to summarize the current status:
* *Reviewer ASAc* (`Score 8`): Highly appreciates our novel method and strong motivation. We have addressed the questions regarding wall-clock time and theoretical bounds.
* *Reviewer o27F* (`Score 6`): Has acknowledged our rebuttal, stating that their major concerns regarding generalization and practical settings are **"well addressed"** and maintained their positive rating.

Regarding *Reviewer 6rr7* (Score 2): We respectfully request the AC to weigh this review carefully, as it appears to be an outlier that overlooks the core contributions of our paper:
1. **Lack of Engagement with New Results:** The reviewer's main request was to compare with "LoRA + Adam". We provided this comparison (Table 3), showing our method achieves competitive performance with lower memory usage (no optimizer states). The reviewer has not acknowledged this critical addition.
2. **Factually Inaccurate Assessment:** The reviewer claimed "no significant improvements," contradicting our empirical evidence showing our method outperforms baselines in 12 out of 18 scenarios across various models.
3. **Ignoring Theoretical Contributions:** Unlike Reviewers ASAc and o27F, Reviewer 6rr7 completely ignored our theoretical framework (Hybrid Smoothness Condition), which is a primary contribution of this work.

Given the **strong endorsement from Reviewer ASAc and the confirmation from Reviewer o27F**, we believe the extreme negative score from **Reviewer 6rr7 does not accurately reflect the quality and contribution of our work**. We hope the AC considers these points during the decision-making process.

Best regards,

Authors

---

### Meta-Review · Area_Chair_2UYK · 2026-01-03

**Summary:**

This paper proposes a new hybrid approach to fine-tuning LLMs, where the LLM and PEFT module are optimized using zeroth-order and first-order optimization methods, respectively. The authors motivate the approach by loss landscapes and analyze it. The approach is evaluated empirically on six tasks, three PEFT methods, and three models. This is a solid paper, with both theory and good experiments, which will motivate more future work. The main concerns of the reviewers are:

* **Limited empirical evaluation:** All experiments use prompt encoder tuning on a sentiment dataset and OPT-1.3b. This was addressed in the rebuttal. The new evaluation is on six tasks, three PEFT methods, and three models.

* **No comparison to LoRA + Adam:** This was addressed in the rebuttal. The proposed method is better in 4 tasks out of 6 (Table 3). Its memory consumption is slightly lower than LoRA + Adam (Table 4).

* **No runtime comparison:** This was addressed in the rebuttal. The proposed method is 4% slower than SGD and 32% slower than LoRA + SGD.

* **Potentially loose theory:** I completely agree. However, this is not just a theory paper. The proposed method is simple, has minimal overhead, and works well.

* **Focus on SGD only:** I agree and assume that Adam is the next natural step. This can be done in future work.

This is a solid paper with a good rebuttal. Therefore, I recommend acceptance.

**Reviewer Concerns:**

**Limited empirical evaluation**, **no comparison to LoRA + Adam**, and **no runtime comparison** were addressed thoroughly in the rebuttal. I do not consider **potentially loose theory** and **focus on SGD only** to be a big problem.

**Reviewer Scores:**

Reviewer o27F maintained their score. Reviewer 6rr7 would likely increased their score to at least 4 because their concerns were addressed. This would put the paper firmly into the acceptance range.

---

### Decision · Program_Chairs · 2026-01-26

Accept (Poster)